

**Chemical characterization and sources of submicron aerosols in the**
**northeastern Qinghai-Tibet Plateau: insights from high-resolution**
**mass spectrometry**
**Xinghua Zhang[1,2,3], Jianzhong Xu[1], Shichang Kang[1], Qi Zhang[4]**
[1]State Key Laboratory of Cryospheric Sciences, Northwest Institute of Eco-Environment and
Resources, Chinese Academy of Sciences, Lanzhou 730000, China
[2]Key Laboratory of Arid Climatic Change and Reducing Disaster of Gansu Province, Key
Laboratory of Arid Climatic Change and Disaster Reduction of CMA, Institute of Arid
Meteorology, China Meteorological Administration, Lanzhou 730020, China
[3]University of Chinese Academy of Sciences, Beijing 100049, China
[4]Department of Environmental Toxicology, University of California, Davis, CA 95616, USA
*Correspondence to*: Jianzhong Xu (jzxu@lzb.ac.cn)
**Abstract**
An Aerodyne high-resolution time-of-flight aerosol mass spectrometer (HR-ToF-AMS) was
deployed along with other online instruments to study the highly time-resolved chemistry and
sources of submicron aerosols (PM$_1$) at Waliguan (WLG) Baseline Observatory, a high-altitude
(3816 m a.s.l.) background station located at the northeastern edge of Qinghai-Tibet Plateau
(QTP), during 1−31 July 2017. The average PM$_1$ mass concentration during this study was 9.1 µg
m$^{-3}$ (ranging from 0.3 to 28.1 µg m$^{-3}$), which was distinct higher than those (2.0–5.7 µg m$^{-3}$)
measured with Aerodyne AMS at other high-elevation sites in the southern or central QTP. Sulfate
showed dominant contribution (38.1%) to PM$_1$ at WLG following by organics (34.5%),
ammonium (15.2%), nitrate (8.1%), BC (3.0%) and chloride (1.1%). Accordingly, bulk aerosols
appeared to be slightly acidic throughout this study mainly related to the enhanced sulfate
contribution. All chemical species peaked at the accumulation mode, indicating the well mixed
and highly aged aerosol particles at WLG from long-range transport. Positive matrix factorization
(PMF) on the high-resolution organic mass spectra resolved four distinct organic aerosol (OA)
components, including a traffic-related hydrocarbon-like OA (HOA), a relatively fresh biomass
burning OA (BBOA), an aged biomass burning OA (agBBOA) and a more-oxidized oxygenated
OA (OOA). On average, the two relatively oxidized OAs, OOA and agBBOA, contributed 34.4%
and 40.4% of organics, respectively, while the rest were 18.4% for BBOA and 6.8% for HOA.
Source analysis for air masses displayed higher mass concentrations of PM$_1$ and enhanced
contributions of sulfate and biomass burning related OA components (agBBOA + BBOA) were
from northeast of the WLG with shorter transport distance, whereas lower PM$_1$ mass
concentrations with enhanced OOA contribution were from west after long-range transport,
suggesting their distinct aerosol sources and significant impacts of regional transport to aerosol
mass loadings and chemistry at WLG.



# 1 Introduction

The Qinghai-Tibet Plateau (QTP) is one of the most remote and pristine region in the world. Its huge surface area (~ 2,500,000 km$^2$) and high elevation (with a mean elevation of more than 4000 m above sea level (a.s.l.)) make it especially important in earth sciences and therefore called as the "third pole" (Yao et al., 2012). According to its high elevation, sparse population and minor local anthropogenic activities, the QTP is regarded as an ideal area for observing the natural background aerosol and long-range transport aerosol. In recent decades, a certain number of studies have presented convincing evidence for the long-range transport of air pollutants from the surrounding areas to the QTP (Engling et al., 2011; Xia et al., 2011; Lüthi et al., 2015; Zhang et al., 2017). Particularly, air pollutants from the southern and southeastern Asia, two of the major regions with enhanced biomass burning emissions in the world, would stack up in the southern foothills of the Himalayas during the pre-monsoon season, then climb over Himalayas by the topographic lifting and the mountain-valley breeze circulation, and finally move upward to QTP (Lüthi et al., 2015). These long-range transport following by deposition of polluted air masses, especially for the two important light-absorbing substances of black carbon (BC) and brown carbon (BrC), have significant impacts on climate, environment and hydrology in the QTP (Xu et al., 2009; Kang et al., 2010; Qian et al., 2011; Yang et al., 2014).

In contrast, aerosol particles in the northern QTP showed quite different behaviors comparing with that in the southern QTP due to the different aerosol sources and climate for these two regions. For example, Li et al. (2016) found equal important contributions from fossil fuel (46%) and biomass (54%) aerosol sources to BC in the Himalayas, however, it was dominated by fossil fuel combustion (66%) in the northern QTP. Correspondingly, the chemical composition of ambient aerosol in the northern QTP was also distinct different with that in the southern QTP. Xu et al. (2014a, 2015) conducted aerosol compositions from filter measurements of PM$_{2.5}$ (particulate matter with diameter less than 2.5 μm) at the Qilian Shan Station observatory at the northeast edge of QTP, and found sulfate was a dominant component during summer season due to the influence of anthropogenic emissions from inland of northwest China. Similar results were also found in Li et al. (2013) and Zhang et al. (2014) which conducted field studies in the northeastern part of QTP. Nitrate, oxidized from NO$_x$, was also an important component in the northern QTP which could interact with mineral dust during transport (Xu et al., 2014a). Due to the relatively lower elevation comparing with the southern QTP (< 4000 vs. > 5000 m a.s.l.), the polluted air masses are easily transported to the mountain areas in the northern QTP forced by the strongly mountain-valley breeze during summer (Xu et al., 2013). In addition, air pollutants to the northern QTP could also from the central Eurasian continent where locates in the upstream of the northwest of China, although relatively lower air masses presented comparing with those impacted by anthropogenic emissions from China and the Indian subcontinent (Xue et al., 2013). However, most of the previous studies in the northeastern QTP for characterizing the chemical properties and sources of aerosol particles were heavily based on the filter or snow/ice samples with low time resolution ranging from days to weeks, mainly because of the absent deployment of real-time instruments at the remote region with harsh environments, challenging weather conditions and logistical difficulties. The real-time measurement of atmospheric aerosol chemistry with high time resolution is still relatively rare in the northern QTP until now.





The Aerodyne aerosol mass spectrometer (AMS) is an unique instrument which can provide both
chemical composition and size distribution information of non-refractory submicron aerosol
(NR-PM$_1$) species with high time resolution and sensitivity (Jayne et al., 2000; Jimenez et al.,
2003; Canagaratna et al., 2007). AMS has been widely implemented worldwide in recent decades,
especially in China since 2006 due to the high attention to atmospheric environment (Li et al.,
2017, and reference therein). Besides the typical applications for studying air pollution in these
urban/rural sites, e.g. megacities with severe haze pollution in eastern China, AMS has also been
successfully deployed at many remote sites in China due to its low detection limitation (see details
in Table 1 of Xu et al. (2018) and Table S1 of Zhang et al. (2018)). In recent years, deployments of
AMS in the highland areas of QTP have been conducted in some field studies, including a
high-resolution time-of-flight AMS (HR-ToF-AMS) and a soot particle AMS (SP-AMS) at Nam
Co in the central QTP (Wang et al., 2017; Xu et al., 2018), a HR-ToF-AMS at QOMS (Zhang et al.,
2018) and Mt. Yulong (Zheng et al., 2017) in the southern QTP, and an Aerodyne aerosol chemical
speciation monitor (ACSM) at Menyuan in the northeastern QTP (Du et al., 2015). Consistent
with those filter samplings, the dominant contributions of organics (54−68%) but low PM$_1$
(NR-PM$_1$ + BC) mass loadings (2.0−5.7 µg m$^{-3}$) were all found in those AMS studies conducted
in the southern or central QTP (Zheng et al., 2017; Xu et al., 2018; Zhang et al., 2018), mainly
associated with the significant impacts of long-range transport biomass burning emissions from
southern Asia, whereas relatively few studies were conducted in the northern QTP.
In this study, a HR-ToF-AMS with other real-time collocated instruments were first deployed at
the Waliguan (WLG) Baseline Observatory, which was one of the World Meteorological
Organization's (WMO) Global Atmospheric Watch (GAW) baseline observatories, located in the
northeastern QTP, to characterize the submicron aerosol chemical compositions and sources
during summer season. The 5-min real-time characterizations of submicron aerosols including
mass concentrations, chemical composition, size distribution as well as temporal and diurnal
variations were presented in details in this study. Source apportionment using positive matrix
factorization (PMF) analysis on the high-resolution mass spectrum of organic aerosol (OA) was
conducted to investigate the sources and chemical evolution of OA during long-range transport.
Finally, back trajectories of air masses were then performed to present the possible sources and
pathway of ambient aerosols during the sampling period.

## 2 Experimental methods

### 2.1 Site and measurements

The field study was carried out during 1−31 July 2017 within the typical warm and rainy season at
the Waliguan (WLG) Baseline Observatory (36°17′ N, 100°54′ E, 3816 m a.s.l.), which locates in
the top of Mt. Waliguan at the northeastern edge of QTP in western China with an ~ 600 m
elevation difference from the surrounding ground (Fig. 1a and b). Mt. Waliguan is a relatively
remote area and generally covered by typical highland vegetation, e.g., highland grassland and
tundra, and constructed as an in-land baseline station of Global Atmosphere Watch (GAW) since
1994 (http://www.wmo.int/pages/prog/arep/gaw/gaw_home_en.html). The closest town, GongHe
County, is located ~ 30 km to the west of Mt. Waliguan and with a population of ~ 30,000, while



Xining, the capital city of Qinghai Province, China, is the closest concentrated population center
located about 90 km to the northeast and with a population of 2.35 millions. A national road is
about 9 km to the north of Mt. Waliguan, yet with relative light vehicle traffic. Therefore, there are
no strong anthropogenic source emissions around Mt. Waliguan. The date and time used in this
study are reported in local time, i.e., Beijing Time (BJT: UTC + 8 h).

### 2.2 Instrumentation

Aerosol particle measurements were performed at the top floor of the main two-story building at
WLG observatory with a suit of real-time instruments, including a HR-ToF-AMS (Aerodyne
Research Inc., Billerica, MA, USA) for 5 min size-resolved chemical compositions (organics,
sulfate, nitrate, ammonium and chloride) of NR-PM$_1$, a photoacoustic extinctiometer (PAX, DMT
Inc., Boulder, CO, USA) for particle light absorption and scattering coefficients ($b_{abs}$ and $b_{scat}$) at
405 nm and the black carbon (BC) mass concentration through a constant mass absorption
efficient (MAE) value of 10.18 m$^2$ g$^{-1}$, and a cloud condensation nuclei counter (CCN-100, DMT
Inc., Boulder, CO, USA) for the number concentration of cloud condensation nuclei (CCN) that
can form into cloud droplets. Simultaneously, other synchronous data were also acquired at the
WLG baseline observatory, including the mass concentrations of PM$_{2.5}$ and PM$_{10}$ measured by a
TEOM 1405-DF dichotomous ambient particulate monitor with a filter dynamics measurement
system (Thermo Scientific, Franklin, MA, USA) and gaseous pollutants of CO and O$_3$ measured
using the Thermo gas analyzers (Model 48i and 49i, respectively, Thermo Scientific, Franklin,
MA, USA). The setup of instruments in this study was shown in Fig. 1d. Ambient particles were
sampled through an inlet system, including a PM$_{2.5}$ cyclone (model URG-2000-30EH, URG Corp.,
Chapel Hill, NC, USA) for removing coarse particles with size cutoffs of 2.5 μm, a nafion dryer
following the cyclone to dry the ambient air and eliminate the potential humidity effect on
particles, and 0.5 inch stainless steel tubes. The inlet stepped out of the building rooftop about 1.5
m, and the total air flow of the inlet was about 12.5 L min$^{-1}$, maintained by a vacuum pump with a
flow rate of 10 L min$^{-1}$ for the PM$_{2.5}$ size cut, and the other part of flow rate by the instruments.
The room temperature was maintained at ~ 18 °C by two air conditioners. In addition, a Vantage
Pro2 weather station (Davis Instruments Corp., Hayward, CA, USA) was set up on the building
rooftop to obtain the real-time meteorology data, including ambient temperature ($T$), relative
humidity (RH), wind speed (WS), wind direction (WD), solar radiation (SR), and precipitation
(Precip.).
The details of the Aerodyne HR-ToF-AMS has been described elsewhere (DeCarlo et al., 2006).
Briefly, a 120 μm critical orifice (replaced the typical 100 μm for enhancing the transmission
efficiency at high-altitude area) and an aerodynamic lens were settled in the front inlet system to
sample and focus the ambient particles into a concentrated and narrow beam. The focused particle
beam exiting the lens was accelerated into the particle-sizing vacuum chamber to obtain the
aerodynamic size of particles by a rotating wheel chopper. Then, particles were vaporized
thermally at ~ 600 °C by a resistively heated surface and ionized by a 70 eV electron impact, and
finally, detected by a high-resolution mass spectrometer. The chopper generally worked at three
positions alternately, i.e., open, close, and chopping positions, for measuring the bulk and
background signals as well as the size-resolved spectral signals of airborne particles, respectively.





In this study, the mass spectrometer was toggled under the high sensitive V-mode (detection limits
~ 10 ng m$^{-3}$) and the high resolution W-mode (~ 6000 m/Δm) every 5 min. Under the V-mode
operation, the instrument also switched between the mass spectrum (MS) mode and the particle
P-ToF mode every 15 s to obtain the mass concentrations and size distributions of NR-PM$_1$
species, respectively, whereas the high resolution W-mode was used to obtain high resolution mass
spectral data.

**2.3 Data processing**

The HR-ToF-AMS data were processed using the standard AMS analysis software of SQUIRREL
(v1.56) to determine the mass concentrations and size distributions of NR-PM$_1$ species and the
high resolution data analysis software of PIKA (v1.15c) to analyze the ion-speciated mass spectra,
components and elemental composition (e.g., oxygen-to-carbon (O/C), hydrogen-to-carbon (H/C),
nitrogen-to-carbon (N/C) and organic mass-to-organic carbon (OM/OC)) of organics in this study.
A collection efficiency (CE) was introduced to compensate for the incomplete transmission and
detection of particles due to particle bouncing at the vaporizer and partial transmission through the
aerodynamic lens. Middlebrook et al. (2012) had evaluated the dependency of CE on several
ambient properties and concluded a composition-dependent CE parameterization according to the
sampling line RH, aerosol acidity, and mass fraction of ammonium nitrate (ANMF). High RH,
high aerosol acidity or high ANMF values would all increase the CE obviously. However, in this
study, (1) aerosol particles were dried totally through a nafion dryer in the inlet system and made
sure that RH in the sampling line were below 40%; (2) aerosol particles were just slightly acidic as
indicated by the average ratio (0.86) of measured ammonium to predicted ammonium (see Sect.
3.1 and Fig. 3a for details); (3) ANMF values were normally below 0.4 during the entire sampling
period as shown in Fig. S1. Therefore, these three parameters were all expected to have negligible
effects on the quantification of aerosol species from our AMS data set and thus a constant CE of 0.5,
which has been widely used in previous field AMS studies, was finally employed in this study. The
source apportionment of organics in this study was conducted by Positive matrix factorization
(PMF) analysis using the PMF2.exe algorithm (v4.2) (Paatero and Tapper, 1994) and PMF
Evaluation Tool (PET, v2.03) (Ulbrich et al., 2009) in robust mode on the high resolution organic
mass spectrum. Note that the data and error matrices input into the PMF analysis were generated
from analyzing the V-mode data via PIKA fitting rather than W-mode in this study due to the low
aerosol mass loading at WLG. The PMF analysis was thoroughly evaluated following the
procedures summarized in Table 1 of Zhang et al. (2011), including modifying the error matrix,
down-weighting or removing the low signal-to-noise ($S/N$) ions. For example, the signals of $H_2O^+$
and $CO^+$ for organics were scaled to that of $CO_2^+$ during this study as $CO^+ = CO_2^+$ and
$H_2O^+ = 0.225 \times CO_2^+$ according to Aiken et al. (2008), while signals of $HO^+$ and $O^+$ were set as
$HO^+ = 0.23 \times H_2O^+$ and $O^+ = 0.04 \times H_2O^+$ based on the fragmentation pattern of water
molecules (Xu et al., 2014b), respectively. Then the above four ions were further down-weighted
by increasing their errors by a factor of 2 in PMF analysis. Isotopic ions were generally excluded
because their signals are not directly measured. The "bad" ions with $S/N < 0.2$ were removed from
the data and error matrices, while the "weak" ions with $0.2 < S/N < 2$ were downweighted by
increasing their errors. In addition, some runs with huge residual spikes, e.g., data with much too
low mass loadings related with the heavy rain on 27 July 2017, were also removed from the data



and error matrices. Finally, a four-factor solution with $f$Peak = 0 was chosen in this study by
examining the model residuals and $Q/Q_{exp}$ contributions for each $m/z$ and time, as well as
comparing the mass spectra of individual factor with reference spectra and the time series of
individual factor with external tracers. The mass spectra, time series, and diurnal variations of
PMF results from three-factor and five-factor solutions were also shown in Fig. S2 and S3 for
comparison, respectively. The three-factor solution did not separate the two biomass burning
factors whereas the five-factor solution showed a splitting factor.

### 3 Results and discussion

### 3.1 Size-resolved chemical characteristics of $PM_1$

An overview of temporal variations of mass concentrations and fractions of $PM_1$ chemical species
(organics, sulfate, nitrate, ammonium, chloride and BC) as well as meteorological conditions ($T$,
RH, WS, WD, and Precip.), mass concentrations of relevant particulate matters ($PM_{2.5}$ and $PM_{10}$)
and gaseous pollutants ($O_3$ and CO), and mass fractions of organic components are shown in Fig.
2, respectively. Missing data are due to hardware or software malfunction, maintenance of the
instrument, or removing large spikes and unique burning event in data processing. Air temperature
($T$) ranged from 8.5 to 14.5 °C for the averaged diurnal variation during the study, with an average
($\pm 1\sigma$) of 11.0 ± 2.0 °C, while relative humidity (RH) ranged from 55.9 to 73.5% with an average
of 66.6 ± 5.7% (Fig. S4). The wind directions (WD) at WLG were predominantly by eastern,
southeastern and northeastern during this study, with an average wind speed (WS) of 4.4 ± 2.8 m
$s^{-1}$ (Fig. 1c and 2b). In addition, WD generally changed from eastern to southeastern during the
nighttime with WS higher than 4 m $s^{-1}$, whereas from northwestern to northeastern during the
daytime with relatively lower WS (Fig. S4). Two moderate rain events occurred during 2−9 and
22−28 July 2017, with daily mean values of 2.6 and 7.4 mm $d^{-3}$, respectively (Fig. 2a).
The total $PM_1$ mass varied dynamically throughout this study with 5-min mass concentration
ranging from 0.3 to 28.1 µg $m^{-3}$. This dynamic variation pattern could also be found for the mass
concentrations of $PM_{2.5}$, $PM_{10}$ and CO, with their correlation coefficients ($R^2$) versus $PM_1$ varying
reasonably from 0.39 to 0.63 (Fig. 2 and S4). In addition, $PM_1$ accounted 66% of $PM_{2.5}$ mass in
this study (Fig. S5), reflecting essentially contribution of submicron aerosols at WLG. Overall,
average mass concentration of total $PM_1$ ($\pm 1\sigma$) at WLG for the entire study was 9.1 ($\pm 5.3$) µg $m^{-3}$,
which was much higher than those at other high-elevation sites in the QTP measured with
Aerodyne AMS, such as 2.0 µg $m^{-3}$ between 31 May and 1 July 2015 at Nam Co Station (4730 m
a.s.l.) in the central of QTP (Xu et al., 2018), 4.4 µg $m^{-3}$ between 12 April and 12 May 2016 at
QOMS (4276 m a.s.l.) at the southern edge of QTP (Zhang et al., 2018), and 5.7 µg $m^{-3}$ between
22 March and 14 April 2015 at Mt. Yulong (3410 m a.s.l.) at the southeastern edge of QTP (Zheng
et al., 2017), whereas this value was comparable with that (11.4 µg $m^{-3}$) measured with an
Aerodyne ACSM between 5 September and 15 October 2013 at Menyuan (3295 m a.s.l.) at the
northeastern QTP (Du et al., 2015). The high mass concentration at WLG was likely due to the
relatively shorter distance from the polluted city center and strongly mountain-valley breeze
during summer. Sulfate and organics were the two dominant $PM_1$ species at WLG, accounting for
38.1% and 34.5% on average, respectively, followed by ammonium (15.2%), nitrate (8.1%), BC



(3.0%) and chloride (1.1%). This chemical composition of $PM_1$ at WLG was quite different with those at Nam Co, QOMS and Mt. Yulong sites in the central or southern QTP (Zheng et al., 2017; Xu et al., 2018; Zhang et al., 2018), where organics was the dominant species accounting for 54−68% of total $PM_1$ mass due to the significant contribution of biomass burning emissions, whereas sulfate only contributed 9−15% of total $PM_1$. The consistent high contribution of sulfate was also observed at Menyuan (28%) in the northeastern QTP and other rural and remote sites (19−64%) in East Asia which were far away from urban areas, as that summarized in Fig. 1 in Du et al. (2015). Moreover, as displayed in Fig. 3b, mass contribution of sulfate increased significantly with the increase of total $PM_1$ mass (lower than 15% for $PM_1$ mass equal to 1.0 µg $m^{-3}$ and increased to more than 45% for $PM_1$ mass of 20.0 µg $m^{-3}$), suggesting important contribution of sulfate to submicron aerosols at WLG.

Bulk acidity of $PM_1$ at WLG was also evaluated according to the method in Zhang et al. (2007), namely using the ratio of measured ammonium to the predicted ammonium that calculated based on the mass concentrations of sulfate, nitrate and chloride and assumed full neutralization of these anions by ammonium. The $PM_1$ appeared to be slightly acidic throughout this study, as indicated by the scatter plot between the measured and predicted ammonium in Fig. 3a (Slope = 0.86, $R^2$ = 0.98). The acidic feature of aerosol particles at WLG was consistent with those results at Menyuan (Du et al., 2015) and Qilian Shan Mountain (Xu et al., 2015) that both located in the northeastern QTP, but different with those at Nam Co (Xu et al., 2018) and QOMS (Zhang et al., 2018) in the central or southern edge of QTP where bulk aerosol particles were generally neutralized or excesses of ammonium. The enriched sulfate in the northeastern QTP might be related tightly with the enhanced coal consumption in the northwest of China and aqueous processing by cloud at the mountains. This conclusion could be further demonstrated by the emission distribution of sulfur dioxide ($SO_2$) in China observed by the OMI satellite instrument in previous studies (Lu et al., 2011; van der A et al., 2017), where $SO_2$ showed considerable concentrations in the northwest of China, especially in urban areas like Xining and Lanzhou cities, whereas extremely low concentrations occurred in the southern QTP.

The average chemically-resolved size distributions of mass concentrations of NR-$PM_1$ species are shown in Fig. 3c. Overall, all chemical species peaked at the accumulation mode with different peaking sizes, e.g. ~ 400 nm in aerodynamic diameter ($D_{va}$) for organics, ~ 450 nm for chloride, and ~ 500 nm for the rest three secondary inorganic species (sulfate, nitrate and ammonium), indicating the well mixed and highly aged aerosol particles at WLG during the sampling period. Moreover, organics presented relatively wider distribution than the three secondary inorganic species in the small sizes (< 300 nm). This could also be clearly revealed by the variations of mass contribution of chemical species as a function of particle sizes in Fig. 3d. The contribution of organics decreased apparently with the increasing sizes whereas those of three inorganic species, especially sulfate, increased correspondingly. Specifically, organics could contributed more than half of the ultrafine NR-$PM_1$ ($D_{va}$ < 100 nm) that maybe associated with the existing of relatively fresh sources of organic particles, while the three inorganic species dominated (more than 60%) at the accumulation mode due to their highly aged properties.

**3.2 Bulk characteristics and elemental composition of OA**



The average high-resolution mass spectrum (HRMS) and elemental compositions of OA during
the study were shown in Fig. 4a. Note that the elemental ratios of O/C, H/C, N/C and OM/OC in
this study were all determined using the "improved-ambient" method (Canagaratna et al., 2015),
which increased O/C by 29%, H/C by 14 % and OM/OC by 15% on average, respectively,
comparing with those determined from the "Aiken ambient" method (Aiken et al., 2008) (Fig. S6).
The average HRMS of OA was quite similar with those at other locations, e.g., Menyuan (Du et al.,
2015), Nam Co (Xu et al., 2018) and QOMS (Zhang et al., 2018) in the QTP, with significantly
high contribution at $m/z$ 44 (composed by $CO_2^+$; 17.9%). On average, $C_xH_yO_1^+$ dominated the
total OA (44.0%) followed by $C_xH_y^+$ (27.9%), $C_xH_yO_2^+$ (21.7%), $H_yO_1^+$ (5.1%), $C_xH_yN_p^+$
(1.0%) and $C_xH_yO_zN_p^+$ (0.2%), as shown in pie chart in Fig. 4a. The total contributions of the two
major oxygenated ion fragments ($C_xH_yO_z^+$) was 65.7% at WLG, which was comparable to those
values at Nam Co during 31 May–1 July 2015 (57.9%; Xu et al., 2018) and QOMS during 12
April–12 May 2016 (66.2%; Zhang et al., 2018), whereas much higher than that (38.0%) measured
during 11 July–7 August 2012 at Lanzhou, an urban city located at the northeastern edge of QTP
(Xu et al., 2014b). In addition, the average O/C ratio of 0.99 in this study was also comparable
with those at Nam Co (0.88; determined by "improved-ambient" method and similarly hereinafter;
Xu et al., 2018) and QOMS (1.07; Zhang et al., 2018), but quite higher than those observed at
various urban and rural sites in China during summertime, e.g., 0.53 and 0.56 in Beijing, 0.40 in
Shanghai, 0.41 in Shenzhen and 0.36 in Jiaxing (Hu et al., 2017). As either the contributions of
$CO_2^+$ and $C_xH_yO_z^+$ or element ratio of O/C are generally considered as good indicators for the
aging degree of OA, the relatively higher values at WLG as well as at other sites in the QTP
together indicated that OA in the QTP was highly oxidized due to the absence of local emissions
and long-range transport.
Diurnal cycles of O/C and OM/OC ratios in this study varied shallowly within 0.96−1.05 and
2.40−2.52, respectively, suggesting an overall regional transport organic aerosol source at WLG
(Fig. 4b). The relatively higher values during afternoon (16:00−17:00 BJT) but lower values
during morning (9:00−10:00 BJT) were mainly related to the different aerosol sources and
photochemical oxidation conditions during long-range transport. Besides, the ratios of O/C and
OM/OC were relatively stable and even higher during nighttime than those in the morning, which
may be induced by the consistent OA source from long-range transport at night whereas relatively
fresh OA enhanced in the morning. Correspondingly, the H/C ratio presented an opposite diurnal
pattern comparing with O/C. The elemental ratios in the Van Krevelen diagram (H/C versus O/C),
which had been used widely to probe the oxidation reaction mechanisms for bulk OA, were
calculated following a slope of –0.64 in this study (Fig. S6), which suggested that the OA
oxidation mechanism at WLG was a combination of carboxylic acid groups with fragmentation
and alcohol/peroxide functional groups without fragmentation (Heald et al., 2010).

### 319    3.3 Source apportionment of OA

PMF analysis on the HRMS of OA identified four distinct components, i.e., a traffic-related
hydrocarbon-like OA (HOA), a relatively fresh biomass burning OA (BBOA), an aged biomass
burning OA (agBBOA) and a more-oxidized oxygenated OA (OOA) in this study. Each of OA
components had unique characteristics on mass spectral profile, average element ratios, diurnal

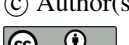


pattern, and temporary variation as well as tight correlations with corresponding tracers. The
details on the source apportionment results of OA are given as follows.
Figure 5 shows the average HRMS and temporal variation of each OA component, respectively. A
traffic-related hydrocarbon-like OA (HOA), with the lowest O/C ratio (0.33) and the highest H/C
ratio (1.83) among the four factors, was identified in this study. Similar to several HOA mass
spectra reported in previous studies, HRMS of HOA in this study was also dominated by
hydrocarbon ion series of $C_nH_{2n\pm1}^+$, especially $C_3H_5^+$ ($m/z$ = 41), $C_3H_7^+$ ($m/z$ = 43), $C_4H_7^+$ ($m/z$ =
55), $C_4H_9^+$ ($m/z$ = 57), $C_5H_9^+$ ($m/z$ = 69), and $C_5H_{11}^+$ ($m/z$ = 71), as shown in Fig. 5a.
Consequently, the dominant contribution of ion fragment was $C_xH_y^+$ (62.8%) follow by $C_xH_yO_1^+$
(29.3%) and $C_xH_yO_2^+$ (6.1%) (Fig. S7), suggesting the primary feature of HOA compared with
other OA components. The two dominant ions, $m/z$ 57 (mainly $C_4H_9^+$ and $C_3H_5O^+$) and $m/z$ 55
(mainly $C_4H_7^+$ and $C_3H_3O^+$), which are generally associated with primary organics from
combustion sources, are commonly considered as tracers for HOA in previous studies (Zhang et
al., 2005). In our study, HOA contributed 71 and 27% to $C_4H_7^+$ and $C_3H_3O^+$, respectively, at $m/z$
while 89 and 29% to $C_4H_9^+$ and $C_3H_5O^+$ at $m/z$ 57. The time series of HOA correlated closely
with those of $C_4H_9^+$ ($R^2$ = 0.68, Fig. 5e) and other alkyl fragments, like $C_3H_7^+$, $C_4H_7^+$, $C_5H_9^+$ ($R^2$
= 0.52−0.65, Fig. S8). Besides, the mass spectrum of HOA was highly similar to those from other
locations around the world (Aiken et al., 2009; Elser et al., 2016; Hu et al., 2016), with correlation
coefficients ($R^2$) varying from 0.62 to 0.94 (Fig. S9). Diurnal variation of HOA (Fig. 6c and d) in
this study presented two slight peaks in the late morning (around 10:00 BJT) and evening (around
20:00 BJT), respectively. Note that the O/C ratio of HOA in this study was obviously higher than
those (generally lower than 0.2) observed in either urban sites or laboratory studies where have
intense local traffic emissions (He et al., 2010; Sun et al., 2011; Xu et al., 2016). The reason is
mainly due to the regional transport of traffic emission to WLG. As mentioned in Sect. 2.1, one
national road is about 9 km to the north of Mt. Waliguan yet with relative light vehicle traffic.
Hence, the traffic related aerosols from either the national road or nearby towns and cities would
undergo certain oxidation processes during transportation to WLG site.
Two biomass burning related OA factors with distinctly different oxidation degrees were also
found in this study. The O/C and OM/OC ratios for the relatively fresh biomass burning OA
(BBOA) were 0.69 and 2.06, respectively, while the aged biomass burning OA (agBBOA) showed
much higher elemental ratios with O/C of 1.02 and OM/OC of 2.49. Correspondingly, the
$C_xH_yO_z^+$ fragment also showed high contribution for agBBOA than that for BBOA (67.8% vs.
56.6%; Fig. S7). Although the $m/z$ 44 (composed by $CO_2^+$) signals were still the highest peaks in
both the two factors, the $m/z$ 60 (composed by $C_2H_4O_2^+$) signals, which were generally regarded
as well-known tracers for biomass burning emissions (Alfarra et al., 2007), was higher in BBOA
than agBBOA HRMS (0.51% vs. 0.46%). In addition, both the fractions of $C_2H_4O_2^+$ in their
HRMS were higher than the typical value of < 0.3% in the absence of biomass burning impacts
(Cubison et al., 2011). As shown in the Fig. 5, the time series of agBBOA correlated tightly with
$C_2H_4O_2^+$ ($R^2$ = 0.79) and sulfate ($R^2$ = 0.47), while BBOA corrected well with $C_2H_4O_2^+$ ($R^2$ =
0.47) and potassium ($R^2$ = 0.30), respectively. The time series of agBBOA also corrected well with
$C_xH_yO_1^+$ and $C_xH_yO_2^+$ ions, while BBOA corrected well with $C_xH_y^+$ and $C_xH_yO_1^+$ (Fig. S8). In
addition, both the mass spectra of the two biomass burning related OA factors resembled well with



that of BBOA at QOMS ($R^2$ of 0.886 and 0.954, respectively; Fig. S9; Zhang et al., 2018),
whereas correlated slightly weaker ($R^2$ = 0.39–0.59) with other standard BBOA mass spectrums at
other sites around the world (Aiken et al., 2009; Mohr et al., 2012). The agBBOA mass spectrum
in this study correlated tightly ($R^2$ = 0.914) with the less oxidized oxygenated OA (LOOOA)
identified at Nam Co station (Fig. S9; Xu et al., 2018). All these comparisons and correlation
analysis further verified the reasonable source apportionment of OA in this study, namely there
were two biomass burning related OAs at WLG which had different oxidation degrees likely due
to their different sources and/or transport distances (see Sect. 3.4 for details). Similar OA source
apportionment of two BBOA components with different oxidation degrees have also been resolved
in previous studies, e.g., an additional oxygenated biomass-burning-influenced organic aerosol
($OOA_2$-BBOA or OOA-BB) in the Paris metropolitan area (Crippa et al., 2013), urban Nanjing
(Zhang et al., 2015) and Mt. Yulong (Zheng et al., 2017), respectively, besides the relatively fresh
BBOA component. Moreover, the O/C ratio of BBOA in this study was also obviously higher than
those in other urban or rural sites in China where had direct or local biomass burning sources, e.g.,
0.24 in Lanzhou (Xu et al., 2016), 0.36 in Beijing (Sun et al., 2016) and 0.26 in Kaiping (Huang et
al., 2011). The diurnal patterns of the two biomass burning related OAs presented nearly opposite
trends in this study (Fig. 6c and d), with high values during the nighttime and decreased trend in
the afternoon for BBOA whereas increased obviously during the daytime for agBBOA, mainly
associated with the possible aging evolution from BBOA to agBBOA via photochemical oxidation
during the daytime.
Another OA component, characterized by the highest peak at $m/z$ 44 (contributed ~ 28% of total
signal and composed by $CO_2^+$), the highest average O/C (1.42) and OM/OC (3.00), and the highest
contribution of $C_xH_yO_z^+$ fragment (44.5% of $C_xH_yO_1^+$ and 30.6% of $C_xH_yO_2^+$; Fig. S7) among
the four factors, was identified as an oxygenated OA (OOA) in this study. The OOA HRMS in this
study was quite similar with those more-oxidized oxygenated OA (MO-OOA) or low-volatility
oxygenated OA (LV-OOA) factors identified frequently in previous AMS studies, especially
resembled tightly to those MO-OOA identified in other QTP locations (Fig. S9), e.g. Nam Co ($R^2$
= 0.995; Xu et al., 2018) and QOMS ($R^2$ = 0.997; Zhang et al., 2018), suggesting that this factor
mainly represented a typical regional oxygenated OA. The time series of OOA in this study
correlated closely with the main secondary inorganic species, sulfate ($R^2$ = 0.51), indicating their
commonly regional and aged properties. In addition, the time series of OOA also corrected well
with $C_xH_yO_2^+$ ions, especially with $CO_2^+$ ($R^2$ = 0.62) as shown in Fig. S8. Although OOA
showed relatively stable contributions during the whole day, the diurnal variation of OOA mass
concentration presented low values in the late morning, continuously increasing trend during the
afternoon and moderate values during nighttime (Fig. 6c and d), suggesting that OOA diurnal
pattern was mainly driven by the combine effects of PBL variation and photochemical activities.
Overall, the average mass concentration of organics was 3.14 µg m$^{-3}$ for the entire study and
composed by 34.4% of OOA, 40.4% of agBBOA, 18.4% of BBOA and 6.8% of HOA on average
(Fig. 6a). The biomass burning related OA components together contributed more than half of the
total organics. In addition, obviously enhanced contributions were found for the two biomass
burning related OA components, particular for agBBOA, with the increasing organics mass,
whereas OOA decreased correspondingly (Fig. 6b). For example, BBOA and agBBOA contributed





only ~ 10% to total organics when OA was less than 1.0 µg m$^{-3}$, whereas the contribution reached
up to 70% with the mass concentration of OA increased to 7 µg m$^{-3}$. Moreover, the important
contribution of agBBOA could also be clearly seen in the temporal variations in Fig. 2f, where
agBBOA dominated organics during the relatively polluted periods. All of these suggested that
biomass burning emissions from regional transport was the important source for OA at WLG. The
triangle plot ($f$44 vs. $f$43 or $f$CO$_2^+$ vs. $f$C$_3$H$_3$O$^+$), which has been widely used in AMS studies, was
an useful method to characterize the possible evolution mechanism of organic components upon
aging in the ambient atmosphere (Ng et al., 2010). As shown in Fig. 4c and d, the majority of data
are distributed within the two dash lines that defined as the general triangular space where ambient
organic components fall by Ng et al. (2010). HOA presented relatively primary nature among four
organic components and located in the bottom of triangle plots, while two biomass related
components in the middle part and OOA in the upper-left corner of the triangle plots, suggesting
an obvious oxidation evolution from relatively primary components to secondary components.

## 3.4 Source analysis

In order to study the dominant sources and explore the influence of regional transport to PM$_1$ mass
loading and chemical composition at WLG during summer season, the 72 h backward air mass
trajectories and average clusters at 500 m above ground level were calculated at 1 h intervals using
the Hybrid Single Particle Lagrangian Integrated Trajectory (HYSPLIT) model (Draxler and
Rolph, 2003) and meteorological data from the NOAA Global Data Assimilation System (GDAS).
Finally, six air mass clusters were adopted in this study as presented in Fig. 7a.
Air masses from northeast (C1) with the shortest transport distance and lowest height among all
the clusters, dominated the air mass contribution (57%) and had the highest average PM$_1$ mass
concentration (10.8 µg m$^{-3}$) during the sampling period, whereas the rest five clusters (C2−C6)
were generally from the west or northwest and showed apparently longer transport distances,
higher heights and relatively lower mass concentrations (5.8−7.8 µg m$^{-3}$) than C1. As shown in
Fig. 1b, three towns (Haiyan, Huangyuan and Huangzhong) as well as the capital city (Xining) of
Qinghai Province were located to the northeast of WLG within 100 km, leading to relatively dense
population and intense industrial activities in these areas compared with those areas to the west of
WLG. Therefore, the prevailing air masses with low transport height for C1 could bring large
amount of surface anthropogenic and industrial pollutants to WLG. This conclusion could further
be supported by the significantly different contributions of chemical species during each cluster
(Fig. 7a). Specifically, C1 showed higher contribution of sulfate compared to other clusters (39.5
vs. 32.0−35.5%), which was mainly related with the intense industrial emissions. In addition, OA
components for C1 showed higher contributions from BBOA (19.5%) and agBBOA (43.3%)
compared with those for C4 and C5 (12.3 and 11.2% for BBOA and 35.4 and 36.7% for agBBOA,
respectively), whereas much lower contribution of oxidized OOA was found for C1 than those for
C4 and C5 (31.0 vs. 43.9 and 44.0%), suggesting the relatively fresh of OA for C1. This
phenomenon was more clear for the two distinct periods, P1 and P2, as shown in Fig. 7b and 8. Air
masses for P2 was mainly from the northeast (C1; 79.0%) and resulted higher contributions from
sulfate (39.9% to total PM$_1$) and the two biomass burning related OA components (BBOA and
agBBOA, 63.2% to total organics), however, three clusters (C4−C6) from the west with long



transport distances dominated P1 and led to significant enhancement of OOA contribution.
Besides the back trajectory analysis, bivariate polar plot analysis was another useful method to
give insight into the potential source regions of ambient aerosols, which presents the relationships
of mass concentrations of PM$_1$ chemical species with wind conditions (WS and WD) (Fig. S10).
All species showed elevated mass concentrations from east, however, with different hotspots for
various species, suggesting their probably distinct sources and impacts from regional transport.
The three main inorganic species (sulfate, nitrate and ammonium) and aged OOA generally had
hotspots from the northeast in accordance with the predominant air masses from northeast during
the daytime where showed more intensive anthropogenic and industrial emissions. Whereas
chloride, BC and BBOA had obvious hotspots from southeast with wind speed around 10 m s$^{-1}$,
which were mainly associated with the possible burning emissions of residents located to the
southeast of WLG during the nighttime.
**4 Conclusions**
In this study, the highly time-resolved physicochemical properties of submicron aerosols were
investigated during summer 2017 at a high-altitude background station in the northeastern QTP,
using a suit of real-time instruments including HR-ToF-AMS, PAX, etc. The major findings
include the following:
1. The 5-min mass concentration of total PM$_1$ (NR-PM$_1$ + BC) varied dynamically between 0.3
and 28.1 µg m$^{-3}$ during this study, with an average PM$_1$ mass loading of 9.1 (± 5.3) µg m$^{-3}$,
which was higher than those measured with Aerodyne AMS at other high-elevation sites in
the southern or central QTP. Different with the significant impacts of biomass burning
emissions in the southern QTP, sulfate showed dominant contribution (38.1%) at WLG. In
addition, mass contribution of sulfate increased obviously with the increase of PM$_1$ mass
loading, indicating the apparently regional transport of sulfate from inland areas in
northwestern China. Correspondingly, PM$_1$ appeared to be slightly acidic throughout this
study related with the enhanced sulfate contribution. All chemical species of NR-PM$_1$ peaked
at the accumulation mode, suggesting the well mixed and highly aged aerosol particles at
WLG during the sampling period.
2. OA on average was dominated by 65.7% of $C_xH_yO_z^+$ ion fragment, with the average O/C
ratio of 0.99 and OM/OC ratio of 2.44, indicating its highly aged property at this remote site.
PMF analysis performed on the OA HRMS resolved four distinct OA components, including
HOA, BBOA, agBBOA and OOA. On average, the two relatively oxidized OAs (OOA and
agBBOA) contributed 34.4% and 40.4%, respectively, while the rest were 18.4% for BBOA
and 6.8% for HOA. In addition, obvious enhanced contributions were found for the two
biomass burning related OA components with the increasing OA mass, demonstrating that
biomass burning emissions from regional transport was the dominant OA source at WLG.
3. Air masses from northeast (C1) with the shortest transport distance among the six clusters
presented dominant contribution (57%) and the highest PM$_1$ mass concentration (10.8 µg
m$^{-3}$), mainly due to the enhanced contributions of sulfate and biomass burning related OA




components from the inland areas in northwestern China. The rest clusters (C2−C6) from the
west or northwest with apparently larger transport distances, however, showed relatively
lower mass concentrations and higher OOA contributions than C1. These source analysis
together suggested the distinct aerosol sources and significant impacts of regional transport
to aerosol mass loadings and chemical compositions at WLG during summer season.
*Data availability.* The processed AMS data and meteorological data in this study are available
upon request from the corresponding author.
*Author contribution.* XHZ analyzed the data and wrote the manuscript. JZX organized the
campaign, analyzed data, and wrote the manuscript. SCK and QZ wrote the manuscript.
*Acknowledgements.* The authors thank the Waliguan Baseline Observatory for the logistical
support with the field campaign and thank the colleagues for continuing support and discussion.
This research was supported by grants from the National Natural Science Foundation of China
(41771079), the Strategic Priority Research Program of Chinese Academy of Sciences, Pan-Third
Pole Environment Study for a Green Silk Road (Pan-TPE) (XDA20040501), and the Chinese
Academy of Sciences Hundred Talents Program.

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





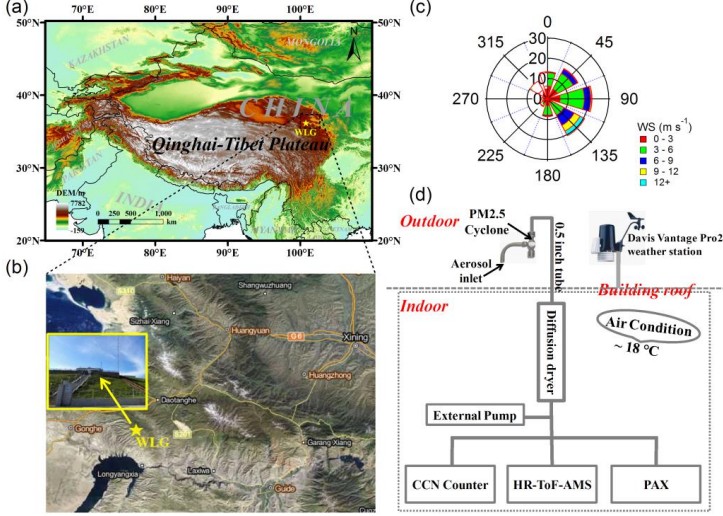


**Figure 1. (a)** Topography map of the Qinghai-Tibet Plateau (QTP), **(b)** location map of Mt. Waliguan Base (WLG; 36.283° N, 100.900° E, 3816 m), **(c)** the wind rose plot colored by wind speed during the field study period, and **(d)** the setup of instruments in this study.

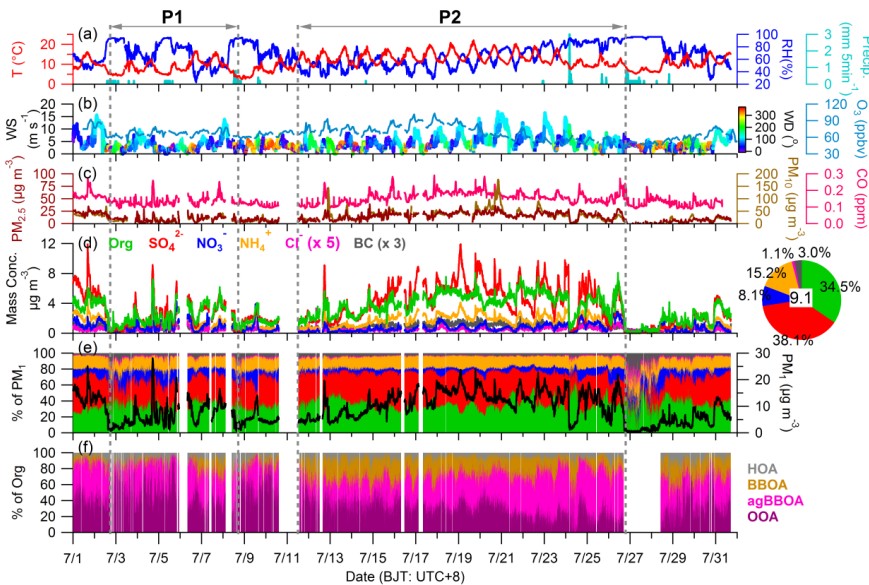


**Figure 2.** Time series of **(a)** ambient temperature ($T$), relative humidity (RH), and precipitation (Precip.), **(b)** wind speed (WS) colored by wind direction (WD) and $O_3$, **(c)** mass concentrations of $PM_{2.5}$, $PM_{10}$, and CO, **(d)** mass concentrations of $PM_1$ species, **(e)** mass contributions of $PM_1$ species as well as the total $PM_1$ mass concentrations, **(f)** mass contributions of four organic components. The pie chart shows the average chemical composition of $PM_1$ for the entire study period, with the average $PM_1$ mass concentration (unit of µg m⁻³) marked in the central.

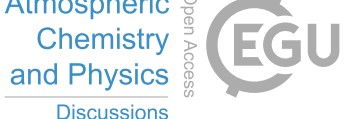



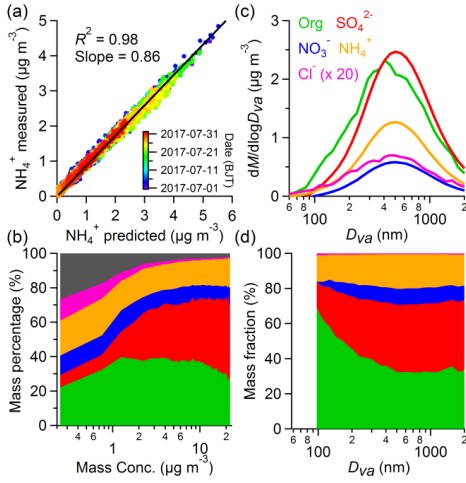

**Figure 3. (a)** Scatterplot and linear regression (black solid line) of measured $NH_4^+$ versus predicted $NH_4^+$ based on the mass concentrations of $SO_4^{2-}$, $NO_3^-$, and $Cl^-$, **(b)** the mass contributions of $PM_1$ chemical species as a function of total $PM_1$ mass concentration, and the average size distributions of **(c)** mass concentrations and **(d)** mass contributions of NR-$PM_1$ species in this study.

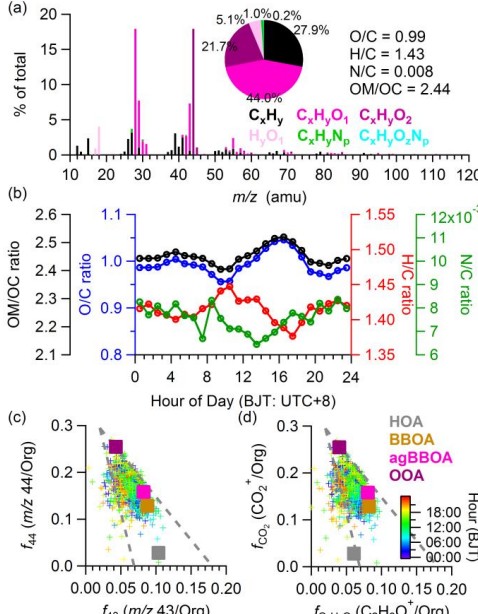

**Figure 4. (a)** The average high-resolution mass spectrum of organics colored with six ion categories (pie charts shows the average contributions of the six ion categories), **(b)** diurnal variations of element ratios (O/C, H/C, N/C, and OM/OC), and scatterplots of **(c)** $f44$ vs. $f43$ and **(d)** $fCO_2^+$ vs. $fC_2H_3O^+$ colored by time of the day, where the corresponding values of four organic components are also shown.





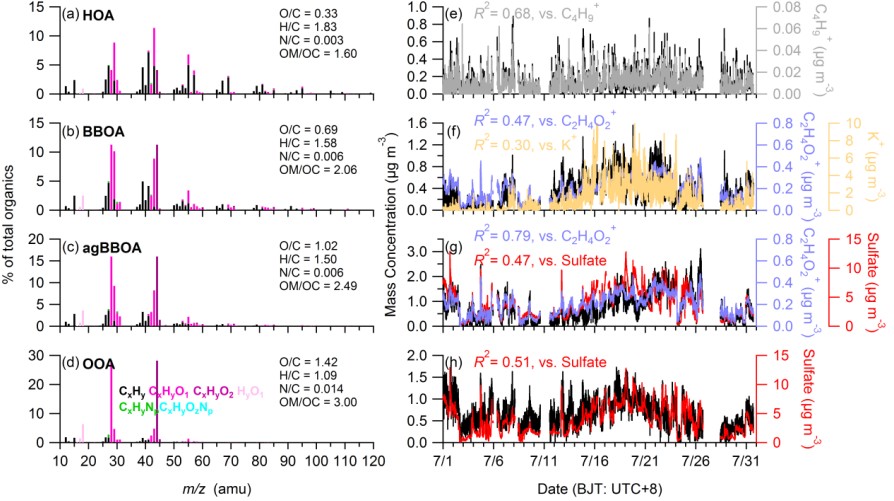

690

**Figure 5.** PMF results of **(left)** high-resolution mass spectra colored by six ion categories for the four OA factors at $m/z < 120$,
**(right)** temporal variations of the four OA factors and corresponding comparison with tracer species.

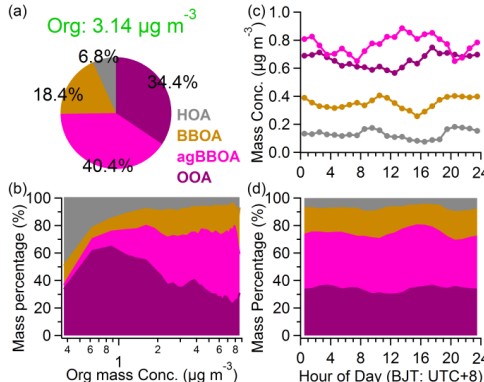

693

**Figure 6.** The average mass contributions of four organic components to total organics **(a)** during the entire study period and **(b)**
as a function of total organics mass concentrations, as well as the diurnal variations of **(c)** mass concentrations and **(d)** mass
contributions of four organic components in this study.





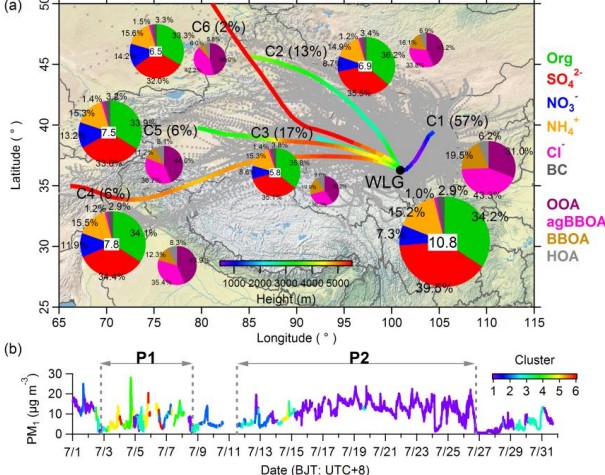

697

**Figure 7. (a)** The 72h backward air mass trajectories (grey dotted lines) and average trajectory clusters (solid lines colored
according to height) calculated at 1 h intervals for the entire study period. Pie charts show the average mass contributions of $PM_1$
species to total $PM_1$ (average $PM_1$ mass are marked in the central of pie charts) and OA components to total organics belong to
each cluster (areas of pie charts are scaled by the corresponding average mass), respectively. **(b)** Temporal variation of $PM_1$ mass
concentration colored by the corresponding cluster name in this study. The markers of P1 and P2 represent two different periods
that selected in this study.

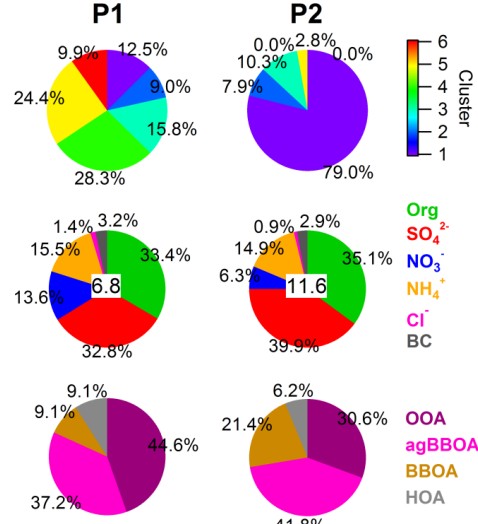

704

**Figure 8. (a)** The occurrence frequency of six air mass trajectory clusters, **(b)** average contributions of $PM_1$ chemical species to
total $PM_1$, and **(c)** average contributions of four organic components to total organics during P1 and P2, respectively.