# Peer review of "Chemical characterization and sources of submicron aerosols in the"

_Atmospheric Chemistry and Physics, 2019_

## Referee Comment (RC1) · Anonymous Referee #1 · 30 Mar 2019

A filed campaign was conducted at Waliguan Baseline Observatory (3816 m a.s.l.), the northeast edge of Qinghai-Tibet Plateau (QTP) during summer season using a high-resolution aerosol mass spectrometer to study the highly time-resolved chemistry and sources of submicron aerosols. The authors found that sulfate dominated the total PM1 in the northern QTP whereas organic aerosols contributed more than half of the total PM1 in the southern or central QTP, suggesting the very different aerosol characteristics and sources in different regions of QTP. Source apportionment of organic aerosol (OA) identified two relatively oxidized OAs, more-oxidized oxygenated OA (OOA) and

aged biomass burning OA (agBBOA). Relatively high mass concentrations of PM1 and enhanced contributions from sulfate and biomass burning related OA components were found for air masses from the northeast of WLG with shorter transport distance, indicating the significant impacts of regional transported aerosols from industrial areas in the northwestern China to high elevation site in the northeastern QTP. Overall, the dataset provided by this work is valuable. The manuscript is overall well written and documented. The topic fits well in the scope of ACP. I recommend this manuscript can be published after some revisions.

Comments:

1. Please specify which method was used for elemental analysis, I-A or A-A?

2. Line 195, 0.23 should be 0.25.

3. The description of geographic orientation in the manuscript need to be checked and revised carefully, especially the usage link north/northern, northeast/northeastern, etc. The right usage should to be "in the northeast edge of QTP" or "in the northeastern QTP" rather than "in the northeastern edge of QTP" or "in the northeast QTP".

4. One of the highlights in this study was the unique aerosol chemical characteristics at WLG compared with other highland sites in the central or southern QTP. Can the author present some direct comparisons via tables or figures besides the simple description in sentences?

5. Line 62-63, "due to the influence of anthropogenic emissions from inland of north-west China". The expression of "inland of northwest China" is incorrect and need to be changed to "...from industrial areas in the northwestern China". Similarly, lines 472 and 488.

6. Line 68, change "strongly" to "strong"

7. Line 73-74, "in the northeastern QTP " moved after "aerosol particles".

8. Line 81, delete "species".

9. Line 96, the expression of "long-range transport biomass burning emissions" was used several times in the whole manuscript, however, it seemed inappropriate and could be changed to "long-range transported biomass burning emissions".

10. Line 117, the expression of "GongHe county" need to be changed to "Gonghe county", consistent with that of "Xining".

11. Line 253-256, this sentence is too long and confusing, please rewritten.

12. Line 274-275, Figure 3c, the scales of y-axis for the size distributions of organics and three inorganic species are not consistent. It is difficult to conclude that "organics presented relatively wider distribution than the three secondary inorganic species in the small sizes".

13. Line 329, "... in previous studies", please add the references.

14. In Figure S9, the author shows the scatter plots of the comparisons between the four high-resolution mass spectra identified in this study and those determined from other studies. Are they HMR or UMR spectra?

15. Line 342-343, please explain more on the diurnal variation of HOA at WLG site.

16. Line 401, the expression of "PBL variation" is inappropriate and needs to be rewritten.

---

## Referee Comment (RC2) · Anonymous Referee #2 · 30 Mar 2019

This paper reports on the chemical characterization and sources of submicron aerosols observed at Waliguan Baseline Observatory, a high-altitude background station in the northeastern Qinghai-Tibet Plateau (QTP), during summer season using a high-resolution time-of-flight aerosol mass spectrometer (HR-ToF-AMS) along with other online instruments. Mass concentrations and fractions of PM1 chemical species, bulk aerosol acidity and size distribution are characterized, respectively. The PM1 mass in the northeastern QTP is obviously higher than those at other high-elevation sites in the southern or central QTP and sulfate dominates the total PM1, which relates tightly with

the regional transport of intense industrial emissions from areas in the northwestern China. Four distinct OA components, including two biomass burning related OAs with different oxidation degree as well as one oxygenated OA and one traffic related OA, are identified by the PMF analysis. Source analysis finally show that the prevailing air masses from northeast with lower transport height and distance can bring surface anthropogenic and industrial pollutants to Mt. Waliguan. Overall, this paper adds new and valuable measurements of aerosol compositions and concentrations in the northeastern QTP, one of the less studied key regions. The paper is within the scope of ACP and generally well written. I recommend publication of this paper in ACP after revisions. I only have some minor points for the author to consider in the revision. Specific Comments: (1) The full expression need to be added when the abbreviations are used first time in the manuscript, such as NOx, PM10, and PBL, etc.. (2) Line 171, does the OM/OC here refer to organic mass or organic matter? (3) Although the AMS mass spectrometer in this study was toggled between V-mode and W-mode every 5 min and the W-mode was used to obtain the high resolution mass spectral data for PMF analysis (in Line 160-165), the author also state in Line 188-190 that the data and error matrices input into the PMF analysis were finally generated from V-mode data rather than W-mode due to the low aerosol mass loading at WLG, hence the entire data used in this study for mass concentration, size distribution and PMF analysis are all from V-mode with 10-min time resolution rather than 5-min, please checked and revised totally. (4) Line 216, when the unique burning event occurred and why was it removed ? (5) Line 239, the author explained the high mass concentration at WLG was due to the relatively shorter distance from the polluted city center and strongly mountain-valley breeze during summer, are there any other evidences to support this conjecture? Such as references, WD variations or air mass trajectories. (6) Line 290, are there any other ion fragments at m/z 44? Please checked carefully for those similar expressions in the whole manuscript. (7) Line 293, "at Lanzhou, an urban city located at the northeastern edge of QTP". The description of Lanzhou that located at the northeast edge of QTP seemed inappropriate. (8) Line 308-313, these sentences for diurnal variations of el-
emental ratios need to be rewritten clearly. Technical Comments: (1) Line 39-41, "Its huge surface area...and therefore called as the third pole". These are two sentences with different subject, please rewritten. (2) The tense in one sentence should consistent, please check the entire manuscript carefully, e.g., Line 48-49, "climb" and "move" need to be changed to "climbed" and "moved", respectively. (3) Line 55, change "that in the southern QTP" to "those in the southern QTP". (4) Line 57, change "however" to "nevertheless". (5) Line 59, change "distinct" to "distinctly". (6) Line 62, change "sulfate was a..." to "sulfate was the". (7) Line 83, change "high attention to" to "great concern of". (8) Line 87, add "the" before "deployment". (9) Line 90-91, change "a HR-ToF-AMS at QOMS (Zhang et al., 2018) and Mt. Yulong (Zheng et al., 2017) in the southern QTP" to "a HR-ToF-AMS at QOMS in the southern QTP (Zhang et al., 2018) and a HR-ToF-AMS at Mt. Yulong in the southeastern QTP (Zheng et al., 2017)". (10) Line 224, "daily mean values" change to "daily mean precipitation" and the unit must be mm d-1 rather than mm d-3. (11) Line 230, add "the" before "average". (12) Line 238, the sentence of "The high mass concentration at WLG..." need to be rewritten, because the mass concentration is just relatively higher when comparing with other sites in the QTP yet much lower than those at other urban or rural sites in China. (13) Line 278, change "contributed" to "contribute". (14) Line 307, change "suggesting an overall regional transport organic aerosol source at WLG" to "suggesting an overall OA source from regional transport at WLG". (15) Line 320, change "HRMS of OA" to "OA HRMS". (16) Line 355, change "high" to "higher" (17) Line 367, "spectrums" change to "spectra".

---

## Referee Comment (RC3) · Anonymous Referee #3 · 9 Apr 2019

This paper presents near real time high resolution aerosol measurements, looking at the chemical composition and sources of organic aerosols. The measurements were taken at a high-altitude background site, northeast of Qinghai-Tibet Plateau, July 2017. The authors found SO4 to dominate PM1 concentrations and identified four different OA sources after PMF analysis. The research presented in this paper will help to better understand OA chemical composition and sources. Overall, the manuscript is well written with a good work on the use of references. The paper, which fits well within the scope of ACP, is recommended to be published after working on the following minor

comments.

Page 2 line 60. Rephrase it, i.e. conducted aerosol composition studies. Page 2 line 63. Rephrase it, i.e.: Similar results were also found by Li et al. Page 2 line 69. Check and rephrase the following paragraph: In addition, air pollutants to the northern QTP could also from the central Eurasian continent where locates in the upstream of the northwest of China, although relatively lower air masses presented comparing with those impacted by anthropogenic emissions from China and the Indian subcontinent (Xue et al., 2013). Page 2 line 77 Rephrase the following paragraph: The real-time measurement of atmospheric aerosol chemistry with high time resolution is still relatively rare in the northern QTP until now Page 3 line 79. Not all AMS instrument provide size distribution. Page 3 line 84. Change to "References therein". Page 3 line 86. Change to "Detection limit". Page 3 line 112. Change to "is located at the top" Page 4 line 125 Change to "Aerosol measurements" or rephrase it. Section 2.2 Instrumentation. I t would be good to add the sampling time of all the instruments. Page 6 lines 202-208. I would move this paragraph to either results or supplement as it is part of results. Page 8 line 283. What is the purpose of comparing these two methods? And/or what is the reason of the increased ratios? Page 9 line 362 Change to "correlated". Page 9 line 362 I would not say that BBOA correlated well with C2H4O2+ with R2 = 0.3.

Technical comments.

A small paragraph about OA sources in these type of sites could added to the introduction. There is not introduction about OA sources while this topic is the focus of this paper. In the mass spectra shown in figure 5, the mass spec agBBOA looks like a semi-volatile OA. More details could be added, perhaps to the supplement, about the analysis mentioned in lines 359-361 to confirm the presence of BBOA, someone would argue you can see a BBOA in summer and it could be more questionable the fact that you are identifying two types of BBOA. A few lines supporting these two BBOA profiles are suggested. The authors can also add, maybe to the supplement, more details on

the way they selected the four factor solution, information about the Q/Qexp values, residuals, etc.

---

## Author Comment (AC1) · 13 May 2019

**Chemical characterization and sources of submicron aerosols in the northeastern Qinghai-Tibet Plateau: insights from high-resolution mass spectrometry**

**Xinghua Zhang et al.**

We appreciate the reviewers for their constructive comments and suggestions. The manuscript has been revised accordingly. Our point-by-point responses to the comments are presented below. The comments are in black, followed by responses in blue and revised manuscript in red with changes marked by underline.

**Response to reviewer #1**

A filed campaign was conducted at Waliguan Baseline Observatory (3816 m a.s.l.), the northeast edge of Qinghai-Tibet Plateau (QTP) during summer season using a high-resolution aerosol mass spectrometer to study the highly time-resolved chemistry and sources of submicron aerosols. The authors found that sulfate dominated the total $PM_1$ in the northern QTP whereas organic aerosols contributed more than half of the total $PM_1$ in the southern or central QTP, suggesting the very different aerosol characteristics and sources in different regions of QTP. Source apportionment of organic aerosol (OA) identified two relatively oxidized OAs, more-oxidized oxygenated OA (OOA) and aged biomass burning OA (agBBOA). Relatively high mass concentrations of $PM_1$ and enhanced contributions from sulfate and biomass burning related OA components were found for air masses from the northeast of WLG with shorter transport distance, indicating the significant impacts of regional transported aerosols from industrial areas in the northwestern China to high elevation site in the northeastern QTP. Overall, the dataset provided by this work is valuable. The manuscript is overall well written and documented. The topic fits well in the scope of ACP. I recommend this manuscript can be published after some revisions.

Thank you very much for your insightful suggestion and positive comments.

**Comments:**

(1) Please specify which method was used for elemental analysis, I-A or A-A?

The elemental ratios in this study were determined using the "improved-ambient" (I-A) method. We have declared this in Sect. 2.3 (Data Processing) and Sect. 3.2 (Bulk characteristics and elemental composition of OA) in the revised version. Specific descriptions are as follows.

"...to analyze the ion-speciated mass spectra, components and elemental compositions (e.g., oxygen-to-carbon (O/C), hydrogen-to-carbon (H/C), nitrogen-to-carbon (N/C) and organic mass-to-organic carbon (OM/OC) ratios using the "improved-ambient" method (Canagaratna et al., 2015)) of organics in this study."

"Note that the elemental ratios of O/C, H/C, N/C and OM/OC in this study were all determined using the "improved-ambient" method (Canagaratna et al., 2015), which increased O/C by 29%, H/C by 14% and OM/OC by 15% on average, respectively, comparing with those determined from the "Aiken ambient" method (Aiken et al., 2008) (Fig. S8)".

(2) Line 195, 0.23 should be 0.25.

Agree. We have changed this value from 0.23 to 0.25 in the revised version. The organic fragmentation in this study was used from the default values. We also rewrote this sentence as follows in the revised version.

"For example, the signals of $H_2O^+$ and $CO^+$ for organics were scaled to that of $CO_2^+$ as $CO^+ = CO_2^+$ and $H_2O^+ = 0.225 \times CO_2^+$, while signals of $HO^+$ and $O^+$ were set as $HO^+ = 0.25 \times H_2O^+$ and $O^+ = 0.04 \times H_2O^+$ according to Aiken et al. (2008)."

(3) The description of geographic orientation in the manuscript need to be checked and revised carefully, especially the usage like north/northern, northeast/northeastern, etc. The right usage should to be "in the northeast edge of QTP" or "in the northeastern QTP" rather than "in the northeastern edge of QTP" or "in the northeast QTP".

Agree. We have checked and revised carefully in the revised manuscript as follows.

"at the northeast edge of Qinghai-Tibet Plateau (QTP)"

"in the northeast part of QTP "

"in the central or southern QTP"

(4) One of the highlights in this study was the unique aerosol chemical characteristics at WLG compared with other highland sites in the central or southern QTP. Can the author present some direct comparisons via tables or figures besides the simple description in sentences?

Thanks for the reviewer's suggestion. A new graph has been added in the supplementary information (Figure S1) to show the comparison of aerosol chemical characteristics among various field studies that conducted at high elevation sites in the QTP. The specific modification in the revised manuscript and Figure S1 added to the supplementary information are shown as follows.

"In recent years, the deployments of AMS in the highland areas of QTP have been conducted in a few field studies (Fig. S1), including..."

[Figure]

**Figure S1.** The field studies conducted at high elevation sites in the Qinghai-Tibet Plateau using AMS or ACSM measurements. The mass concentrations of $PM_1$ and the mass contributions of each chemical species (pie chart) are presented in each site.

(5) Line 62-63, "due to the influence of anthropogenic emissions from inland of northwest China". The expression of "inland of northwest China" is incorrect and need to be changed to "...from industrial areas in the northwestern China". Similarly, lines 472 and 488.

Agree. We have made changes to the three lines in the revised manuscript as follows.

"...due to the influence of anthropogenic emissions from industrial areas in the northwestern China"

"...indicating the apparently regional transport of sulfate from industrial areas in the northwestern China"

"...mainly due to the enhanced contributions of sulfate and biomass burning related OA components from the industrial areas in the northwestern China"

(6) Line 68, change "strongly" to "strong".

Corrected.

(7) Line 73-74, "in the northeastern QTP " moved after "aerosol particles".

Corrected.

(8) Line 81, delete "species".

Corrected.

(9) Line 96, the expression of "long-range transport biomass burning emissions" was used several times in the whole manuscript, however, it seemed inappropriate and could be changed to "long-range transported biomass burning emissions".

Thanks a lot for the reviewer's suggestion. The expressions of "long-range transport" are used several times in the whole manuscript, however, only those used as adjective are changed to "long-range transported", whereas those used as noun are still as "long-range transport" in the revised version. For example,

"...for observing the natural background aerosol and long-range transported aerosol"

"...indicating the well mixed and highly aged aerosol particles at WLG from long-range transport"

(10) Line 117, the expression of "GongHe county" need to be changed to "Gonghe county", consistent with that of "Xining".

Corrected.

(11) Line 253-256, this sentence is too long and confusing, please rewritten.

We have rewritten this long sentence into two short sentences in the revised version as follow.

"Bulk acidity of $PM_1$ at WLG was also evaluated according to the method in Zhang et al. (2007). The predicted ammonium was calculated based on the mass concentrations of sulfate, nitrate and chloride and assumed full neutralization of these anions by ammonium."

(12) Line 274-275, Figure 3c, the scales of y-axis for the size distributions of organics and three inorganic species are not consistent. It is difficult to conclude that "organics presented relatively wider distribution than the three secondary inorganic species in the small sizes".

Thanks for the reviewer's insightful suggestion.

We have modified Figure 3c in the revised version. In Figure 3c, it is clear that OA present a relatively wider distribution than the other three secondary inorganic species in the small sizes (< 300 nm).

[Figure]

**Figure 3. (a)** Scatterplot and linear regression (black solid line) of measured $NH_4^+$ versus predicted $NH_4^+$ based on the mass concentrations of $SO_4^{2-}$, $NO_3^-$, and $Cl^-$, **(b)** the mass contributions of $PM_1$ chemical species as a function of total $PM_1$ mass concentration, and the average size distributions of **(c)** mass concentrations and **(d)** mass contributions of NR-$PM_1$ species in this study.

(13) Line 329, "... in previous studies", please add the references.

We have added the corresponding references in the revised version as follows.

"Similar to several HOA mass spectra reported in previous studies (Zhang et al., 2005; Ng et al., 2011), HRMS of HOA in this study was also dominated by hydrocarbon ion series..."

(14) In Figure S9, the author shows the scatter plots of the comparisons between the four high-resolution mass spectra identified in this study and those determined from other studies. Are they HMR or UMR spectra?

All the mass spectra in Figure S11 in the revised version (corresponding to Figure S9 in previous version) are HMR. We have clarified this point in the manuscript and supplementary information, respectively.

"Besides, the high-resolution mass spectrum of HOA was highly similar to those from other locations around the world..."

"Figure S11. Scatter plots of the comparisons between the four high-resolution mass spectra identified in this study and those high-resolution mass spectra determined from other studies."

(15) Line 342-343, please explain more on the diurnal variation of HOA at WLG site.

We have added a specific explanation on the diurnal variation of HOA in the revised version as follows.

"Although there was not traffic rush hour in the high-elevation site, the increasing vehicles on the national road combined with the valley breeze together lead to the slightly higher HOA concentrations in the late morning, then HOA decreased continuously with the increasing planetary boundary layer (PBL) height in the afternoon and elevated again to a stable high level during the nighttime due to the low PBL height and mountain breeze."

(16) Line 401, the expression of "PBL variation" is inappropriate and needs to be rewritten.

We have rewritten this sentence as follows.

"Although OOA showed relatively stable contributions throughout the whole day, the OOA mass concentrations also presented distinct diurnal variation at WLG site, namely relatively low values in the late morning, continuously increasing trend during the afternoon and moderate values at the nighttime (Fig. 6c and d), which was tightly associated with the photochemical activities in the daytime, aqueous-processing of OA at nighttime as well as the diurnal variation of PBL height."

**Response to reviewer #2**

This paper reports on the chemical characterization and sources of submicron aerosols observed at Waliguan Baseline Observatory, a high-altitude background station in the northeastern Qinghai-Tibet Plateau (QTP), during summer season using a high-resolution time-of-flight aerosol mass spectrometer (HR-ToF-AMS) along with other online instruments. Mass concentrations and fractions of $PM_1$ chemical species, bulk aerosol acidity and size distribution are characterized, respectively. The $PM_1$ mass in the northeastern QTP is obviously higher than those at other high-elevation sites in the southern or central QTP and sulfate dominates the total $PM_1$, which relates tightly with the regional transport of intense industrial emissions from areas in the northwestern China. Four distinct OA components, including two biomass burning related OAs with different oxidation degree as well as one oxygenated OA and one traffic related OA, are identified by the PMF analysis. Source analysis finally show that the prevailing air masses from northeast with lower transport height and distance can bring surface anthropogenic and industrial pollutants to Mt. Waliguan. Overall, this paper adds new and valuable measurements of aerosol compositions and concentrations in the northeastern QTP, one of the less studied key regions. The paper is within the scope of ACP and generally well written. I recommend publication of this paper in ACP after revisions. I only have some minor points for the author to consider in the revision.

We thank the reviewer for his/her careful review of the manuscript.

**Comments:**

(1) The full expression need to be added when the abbreviations are used first time in the manuscript, such as $NO_x$, $PM_{10}$, and PBL, etc..

Thanks for the reviewer's suggestion. We have checked the manuscript carefully and added the full expressions to all of the abbreviations when they are used at the first time as follows.

"Nitrate, oxidized from the nitric oxides ($NO_x$), was also an important component in the northern QTP..."

"...including the mass concentrations of $PM_{2.5}$ and $PM_{10}$ (particulate matter with diameter less than 10 μm) measured by a TEOM 1405-DF dichotomous ambient particulate monitor with a filter dynamics measurement system (Thermo Scientific, Franklin, MA, USA) and gaseous pollutants of carbon monoxide (CO) and ozone ($O_3$) measured using the Thermo gas analyzers..."

"This conclusion could be further demonstrated by the emission distribution of sulfur dioxide ($SO_2$) in China observed by the Ozone Monitoring Instrument (OMI) satellite data in previous studies"

"...then HOA decreased continuously with the increasing planetary boundary layer (PBL) height in the afternoon..."

(2) Line 171, does the OM/OC here refer to organic mass or organic matter?

The OM/OC in this study refer to the ratio of organic mass to organic carbon. The definition of OM/OC added in the revised version is shown as follows.

"to analyze the ion-speciated mass spectra, components and elemental compositions (e.g., oxygen-to-carbon (O/C), hydrogen-to-carbon (H/C), nitrogen-to-carbon (N/C) and organic mass-to-organic carbon (OM/OC) ratios using the "improved-ambient" method (Canagaratna et al., 2015)) of organics in this study."

(3) Although the AMS mass spectrometer in this study was toggled between V-mode and W-mode every 5 min and the W-mode was used to obtain the high resolution mass spectral data for PMF analysis (in Line 160-165), the author also state in Line 188-190 that the data and error matrices input into the PMF analysis were finally generated from V-mode data rather than W-mode due to the low aerosol mass loading at WLG, hence the entire data used in this study for mass concentration, size distribution and PMF analysis are all from V-mode with 10-min time resolution rather than 5-min, please checked and revised totally.

We thank the reviewer for your insightful suggestion. The data time resolution in this study is indeed 10-min rather than 5-min because we do not use the W-mode data due to the low aerosol mass loading in the QTP. We have rewritten this sentence as follows.

"Similar to most of the previous AMS field measurements, the mass spectrometer was toggled under the high sensitive V-mode (detection limits $\sim$ 10 ng m$^{-3}$) and the high resolution W-mode ($\sim$ 6000 m/$\Delta$m) every 5 min in this study. Under the V-mode operation, the instrument also switched between the mass spectrum (MS) mode and the particle P-ToF mode every 15 s to obtain the mass concentrations and size distributions of NR-PM$_1$ species, respectively, whereas the high resolution W-mode was used to obtain high resolution mass spectral data. However, the data and error matrices inputted into the PMF analysis were finally generated from the V-mode data rather than the W-mode data in this study due to the low aerosol mass loading at WLG. Hence, all the data used in this study are from V-mode with 10 min time resolution."

(4) Line 216, when the unique burning event occurred and why was it removed?

The unique burning event mentioned in this study was a local Tibetan festival event occurred during 5−6 July 2017. During the festival, hundreds of local Tibetans gathered together from the evening of 5 July till the late morning of next day to worship their religious god by burning large amounts of specific biofuels when chanting around the pagoda. The emissions from these hundreds of vehicles and motorcycles and biomass burning activities together led to an extraordinary air pollution condition around the study site. Intense burst of aerosol emissions occurred at the midnight on 5 July, with the mass concentration of PM$_1$ increased significantly from less than 10 µg m$^{-3}$ at 23:00 BJT on 5 July to the maximum value of 232 µg m$^{-3}$ at 01:30 BJT on 6 July. Hence, the extremely high aerosol mass loadings during the event can impact on the average results of field measurement and need to be removed. We also have added a specific introduction to this unique burning event in the revised version as follow.

"The missing data are due to hardware or software malfunction, maintenance of the instrument, or removing large spikes and unique burning event (a local Tibetan festival event occurred during 5−6 July 2017 with extremely high aerosol mass loadings) in data processing."

(5) Line 239, the author explained the high mass concentration at WLG was due to the relatively shorter distance from the polluted city center and strongly mountain-valley breeze during summer,

are there any other evidences to support this conjecture? Such as references, WD variations or air mass trajectories.

As mentioned in the manuscript, the $PM_1$ mass concentration at WLG (9.1 µg m$^{-3}$) in the northeastern QTP was much higher than those at NamCo (2.0 µg m$^{-3}$) in the central QTP and QOMS (4.4 µg m$^{-3}$) in the southern QTP. We compared the air mass trajectories at WLG during July 2017 in this study with those at NamCo during June 2015 in Xu et al. (2018) and QOMS during April-May 2016 in Zhang et al. (2018) (Figure R1). Clearly predominant northeastern winds (57%) with shorter transported distance and lower transported height were found at WLG in this study, whereas long-range transported air masses from south Asia dominated apparently at Nam Co Station and QOMS. We have added a specific explanation in the revised manuscript as follow.

"The higher $PM_1$ mass concentration at WLG in the northeastern QTP comparing with those at other sites in the central or southern QTP was likely due to the relatively shorter distance from the industrial areas (e.g., Xining city) in the northwestern China and strong mountain-valley breeze during summer. This conclusion could be supported by the comparisons of air mass back-trajectories between WLG in this study (see Sect. 3.4 for details) and those at Nam Co Station in Xu et al. (2018) and QOMS in Zhang et al. (2018)."

[Figure]

**Figure R1.** Air mass back trajectories at WLG in this study comparing with those at Nam Co Station (NCOS) in Xu et al. (2018) and QOMS in Zhang et al. (2018).

(6) Line 290, are there any other ion fragments at m/z 44? Please checked carefully for those similar expressions in the whole manuscript.

We have checked carefully according to the reviewer's suggestion. The ion at m/z 44 is only $CO_2^+$ in this study and only $C_2H_4O_2^+$ at m/z 60. We have rewritten those corresponding sentences in the revised version.

"...with significantly high contribution at *m/z* 44 (17.9%; composed totally by $CO_2^+$ in this study and similarly hereinafter)"

(7) Line 293, "at Lanzhou, an urban city located at the northeastern edge of QTP". The description of Lanzhou that located at the northeast edge of QTP seemed inappropriate.

We have deleted this description.

(8) Line 308-313, these sentences for diurnal variations of elemental ratios need to be rewritten clearly.

Thanks for the reviewer's suggestion. We have rewritten these sentences in the revised version. Specific changes are as follows.

"The relatively higher O/C and OM/OC during afternoon potentially related with the photochemical oxidation processes in the daytime, while lower values in the late morning mainly associated with the transport of relatively fresh OA from nearby areas to WLG site, which could be further revealed by the corresponding higher H/C and N/C ratios in the late morning as well as the diurnal variations of the two primary OA components (see Sect. 3.3 for details)."

**Technical Comments:**

(1) Line 39-41, "Its huge surface area...and therefore called as the third pole". These are two sentences with different subject, please rewritten.

Agree. We have made changes to this sentence in the revised version.

"Its huge surface area (~ 2,500,000 km$^2$) and high elevation (with a mean elevation of more than 4000 m above sea level (a.s.l.)) make it especially important in earth sciences and therefore the QTP is generally called as the "third pole" (Yao et al., 2012)".

(2) The tense in one sentence should consistent, please check the entire manuscript carefully, e.g., Line 48-49, "climb" and "move" need to be changed to "climbed" and "moved", respectively.

Thanks for the reviewer's suggestion. We have checked the entire manuscript carefully to make the tense consistent in the manuscript.

(3) Line 55, change "that in the southern QTP" to "those in the southern QTP".

Corrected.

"...aerosol particles in the northern QTP showed quite different behaviors comparing with those in the southern QTP..."

(4) Line 57, change "however" to "nevertheless".

Agree.

"For example, Li et al. (2016) found equal important contributions from fossil fuel (46%) and biomass (54%) aerosol sources to BC in the Himalayas, nevertheless, it was dominated by fossil fuel combustion (66%) in the northern QTP."

(5) Line 59, change "distinct" to "distinctly".

We have checked the entire manuscript carefully to make sure the expressions like this are used correctly.

"Correspondingly, the chemical composition of ambient aerosol in the northern QTP was also distinctly different with that in the southern QTP."

(6) Line 62, change "sulfate was a..." to "sulfate was the".

Agree.

"...and found sulfate was the dominant component during summer season..."

(7) Line 83, change "high attention to" to "great concern of".

Corrected.

"AMS has been widely implemented worldwide in recent decades, especially in China since 2006 due to the great concern of atmospheric environment"

(8) Line 87, add "the" before "deployment".

Corrected.

"In recent years, the deployments of AMS in the highland areas of QTP have been conducted in various field studies..."

(9) Line 90-91, change "a HR-ToF-AMS at QOMS (Zhang et al., 2018) and Mt. Yulong (Zheng et al., 2017) in the southern QTP" to "a HR-ToF-AMS at QOMS in the southern QTP (Zhang et al., 2018) and a HR-ToF-AMS at Mt. Yulong in the southeastern QTP (Zheng et al., 2017)".

Thanks for the suggestion. We have made changes as following in the revised version.

"a HR-ToF-AMS at QOMS in the southern QTP (Zhang et al., 2018) and a HR-ToF-AMS at Mt. Yulong in the southeastern QTP (Zheng et al., 2017)"

(10) Line 224, "daily mean values" change to "daily mean precipitation" and the unit must be mm $d^{-1}$ rather than mm $d^{-3}$.

We made a mistake here and revised it as follows.

"...with daily mean precipitation of 2.6 and 7.4 $mm\ d^{-1}$, respectively "

(11) Line 230, add "the" before "average".

Corrected.

"Overall, the average mass concentration of total $PM_1$ (± 1σ) at WLG for the entire study was 9.1 (± 5.3) $\mu g\ m^{-3}$..."

(12) Line 238, the sentence of "The high mass concentration at WLG..." need to be rewritten, because the mass concentration is just relatively higher when comparing with other sites in the QTP yet much lower than those at other urban or rural sites in China.

Thanks a lot for the reviewer's suggestion. We have rewritten this sentence clearly to avoid ambiguity.

"The higher $PM_1$ mass concentration at WLG in the northeastern QTP comparing with those at other sites in the central or southern QTP was likely due to..."

(13) Line 278, change "contributed" to "contribute".

Corrected.

"Specifically, organics could contribute more than half of the ultrafine NR-$PM_1$..."

(14) Line 307, change "suggesting an overall regional transport organic aerosol source at WLG" to "suggesting an overall OA source from regional transport at WLG".

Corrected.

(15) Line 320, change "HRMS of OA" to "OA HRMS".

Corrected.

"PMF analysis on the OA HRMS identified four distinct components..."

(16) Line 355, change "high" to "higher".

Corrected.

"Correspondingly, the $C_xH_yO_z^+$ fragment also showed higher contribution for agBBOA than that for BBOA..."

(17) Line 367, "spectrums" change to "spectra".

Corrected.

"...with other standard BBOA mass spectra at other sites around the world..."

**Response to reviewer #3**

This paper presents near real time high resolution aerosol measurements, looking at the chemical composition and sources of organic aerosols. The measurements were taken at a high-altitude background site, northeast of Qinghai-Tibet Plateau, July 2017. The authors found $SO_4$ to dominate $PM_1$ concentrations and identified four different OA sources after PMF analysis. The research presented in this paper will help to better understand OA chemical composition and sources. Overall, the manuscript is well written with a good work on the use of references. The paper, which fits well within the scope of ACP, is recommended to be published after working on the following minor comments.

Thank you very much for your insightful suggestion and positive comments.

**Comments:**

(1) Page 2 line 60. Rephrase it, i.e. conducted aerosol composition studies.

We have rephrased this sentence according to the reviewer's comment.

"Xu et al. (2014a, 2015) conducted aerosol composition studies from filter measurements..."

(2) Page 2 line 63. Rephrase it, i.e.: Similar results were also found by Li et al.

Agree. We have changed "found in Li et al." to "found by Li et al." in the revised version.

"Similar results were also found by Li et al. (2013) and Zhang et al. (2014) which conducted field studies in the northeast part of QTP."

(3) Page 2 line 69. Check and rephrase the following paragraph: In addition, air pollutants to the northern QTP could also from the central Eurasian continent where locates in the upstream of the northwest of China, although relatively lower air masses presented comparing with those impacted by anthropogenic emissions from China and the Indian subcontinent (Xue et al., 2013).

We have rewritten this sentence as follows.

"Besides the significant impacts by anthropogenic emissions from the northwestern China or Indian subcontinent, air pollutants to the northeastern QTP could also from the central Eurasian continent (Xue et al., 2013)."

(4) Page 2 line 77. Rephrase the following paragraph: The real-time measurement of atmospheric aerosol chemistry with high time resolution is still relatively rare in the northern QTP until now.

We have rephrased the sentence as following according to the reviewer's comment.

"Studies focused on the atmospheric aerosol chemical compositions in the northeastern QTP using the high-time-resolution real-time measurements are still relatively rare until now."

(5) Page 3 line 79. Not all AMS instrument provide size distribution.

We have rewritten this sentence in the revised version.

"The Aerodyne aerosol mass spectrometer (AMS) is a unique instrument which can provide both chemical composition and/or size distribution information of non-refractory submicron aerosol

(NR-PM$_1$) with high time resolution and sensitivity (Jayne et al., 2000; Jimenez et al., 2003; Canagaratna et al., 2007)."

(6) Page 3 line 84. Change to "References therein".

Corrected.

"AMS has been widely implemented worldwide in recent decades, especially in China since 2006 due to the great concern of atmospheric environment (Li et al., 2017, and references therein)."

(7) Page 3 line 86. Change to "Detection limit".

Corrected.

"...AMS has also been successfully deployed at many remote sites due to its low detection limit (Fig. S1)."

(8) Page 3 line 112. Change to "is located at the top"

We have changed "which locates in the top" to "which is located at the top" in the sentence.

"...which is located at the top of Mt. Waliguan "

(9) Page 4 line 125 Change to "Aerosol measurements" or rephrase it.

We have changed "Aerosol particle measurements" to "Aerosol measurements" in the revised version.

"Aerosol measurements were performed at "

(10) Section 2.2 Instrumentation. It would be good to add the sampling time of all the instruments.

Thanks for the suggestion. All the data in this study measured by HR-ToF-AMS, SMPS, CCN or other synchronous instruments are all from 1 to 31 July 2017. We have rephrased the sentences in the "Section 2.2 Instrumentation" clearly in the revised version.

"Aerosol measurements were performed at the top floor of the main two-story building at WLG observatory from 1 to 31 July 2017 with a suit of real-time instruments, including..."

"Simultaneously, other synchronous data were also acquired at the WLG baseline observatory during the sampling period, including..."

(11) Page 6 lines 202-208. I would move this paragraph to either results or supplement as it is part of results.

Thanks for the suggestion. In our opinion, this paragraph is still belonging to the method part about PMF analysis in "Sect. 2.3 Data processing". The paragraph is used to detail the key diagnostic plots of PMF results for this study, and to illustrate the reason why we selected the four-factor solution finally via examining the model residuals, scaled residuals, Q/Q$_{exp}$ contributions for each m/z and time, and other factors. Hence, this paragraph is better to be placed in the method part.

"A summary of the key diagnostic plots of PMF results for this study is presented in Fig. S3. Overall, the PMF solutions were investigated from one to six factors with the rotational parameter ($f$Peak) varying from −1 to 1 with a step of 0.1. Finally, a four-factor solution with $f$Peak = 0 was chosen in this study by examining the model residuals, scaled residuals and Q/Q$_{exp}$ contributions for each $m/z$ and time, as well as comparing the mass spectra of individual factor with reference spectra and the time series of individual factor with external tracers. The mass spectra, time series, and diurnal variations of PMF results from three-factor and five-factor solutions were also shown

in Fig. S4 and S5 for comparison, respectively. The three-factor solution did not separate the two biomass burning factors whereas the five-factor solution showed a splitting factor."

(12) Page 8 line 283. What is the purpose of comparing these two methods? And/or what is the reason of the increased ratios?

The Aerodyne HR-ToF-AMS is widely used to measure the OA elemental composition which can provide useful constraints for understanding aerosol sources, processes, impacts, and fate, and for experimentally constraining and developing predictive aerosol models on local, regional, and global scales (Canagaratna al., 2015). For the previous AMS studies commonly using "Aiken-Ambient" method, the $H_2O^+$ and $CO^+$ ion intensities was empirically estimated rather than directly measured to avoid the gas phase air interferences from gaseous $N_2$ and $H_2O$, namely the $H_2O^+/CO_2^+$ and $CO^+/CO_2^+$ ratios were empirically estimated from limited ambient OA measurements available at the time to be 0.225 and 1, respectively (Aiken et al., 2008; Canagaratna al., 2015). However, this method produced larger biases for alcohols and simple diacids (Canagaratna al., 2015). A detailed examination of the $H_2O^+$, $CO^+$, and $CO_2^+$ fragments in the high-resolution mass spectra of the standard compounds indicates that the "Aiken-Ambient" method underestimates the $CO^+$ and especially $H_2O^+$ produced from many oxidized species (Canagaratna al., 2015). The "Improved-Ambient" method performed by Canagaratna al. (2015) used specific ion fragments as markers to correct for molecular functionality-dependent systematic biases, namely $O/C_{I-A} = O/C_{A-A} \times (1.26 - 0.623 \times f_{CO2+} + 2.28 \times f_{CHO+})$ and $H/C_{I-A} = H/C_{A-A} \times (1.07 + 1.07 \times f_{CHO+})$. The Improved-Ambient elemental ratios are expressed as a product of Aiken-Ambient elemental ratios and a composition-dependent correction factor, which allows for simple recalculation of the Improved-Ambient elemental ratios from Aiken-Ambient values without the need for performing a re-analysis of the raw mass spectra and can be easily applied to already published AMS results (Canagaratna al., 2015).

Therefore, it was quite important to present both the elemental ratios from the two methods in recent AMS studies, which can be easily used to compare with those ratios from either previous AMS studies using "Aiken-Ambient" method or recent AMS studies using "Improved-Ambient" method. Besides, as mentioned above, the increased ratios using the "Improved-Ambient" method are mainly due to the underestimation of $CO^+$ and $H_2O^+$ using "Aiken-Ambient" method.

(13) Page 9 line 362. Change to "correlated".

Agree. We also checked the entire manuscript carefully and corrected the similar issue at other sentences.

"As shown in the Fig. 5, the time series of agBBOA correlated tightly with $C_2H_4O_2^+$ ($R^2 = 0.79$) and sulfate ($R^2 = 0.47$), while BBOA correlated slightly weak with $C_2H_4O_2^+$ ($R^2 = 0.47$) and potassium ($R^2 = 0.30$), respectively. The time series of agBBOA also correlated well with $C_xH_yO_1^+$ and $C_xH_yO_2^+$ ions, while BBOA correlated well with $C_xH_y^+$ and $C_xH_yO_1^+$ (Fig. S9)."
"In addition, the time series of OOA also correlated well with..."

(14) Page 9 line 362. I would not say that BBOA correlated well with $C_2H_4O_2^+$ with $R^2 = 0.3$.

Thanks for the reviewer's suggestion. We have rephrased this sentence in the revised version.

"As shown in the Fig. 5, the time series of agBBOA correlated tightly with $C_2H_4O_2^+$ ($R^2 = 0.79$) and sulfate ($R^2 = 0.47$), while BBOA correlated slightly weak with $C_2H_4O_2^+$ ($R^2 = 0.47$) and potassium ($R^2 = 0.30$), respectively."

**Technical comments.**

(1) A small paragraph about OA sources in these type of sites could be added to the introduction. There is not introduction about OA sources while this topic is the focus of this paper.

We have added a small paragraph about the OA sources in the high-altitude sites in the QTP to Introduction section in the revised version as follows.

"In addition to the low $PM_1$ (NR-$PM_1$ + BC) mass loadings, the dominant contribution from organic aerosol (OA) (54−68%) was found in the southern and central QTP (Zheng et al., 2017; Wang et al., 2017; Xu et al., 2018; Zhang et al., 2018). OA was composed by oxygenated OA (OOA) and biomass burning related OA (BBOA) components in those high-altitude background sites. The OOA component was associated with the intense oxidation processes that converted fresh OA to secondary OA, while BBOA was related to the direct emissions from the biomass burning activities in the highland areas. However, relatively few studies have been conducted in the northern QTP except a measurement using Aerodyne aerosol chemical speciation monitor (ACSM) at Menyuan (Du et al., 2015)."

(2) In the mass spectra shown in figure 5, the mass spec agBBOA looks like a semi-volatile OA. More details could be added, perhaps to the supplement, about the analysis mentioned in lines 359-361 to confirm the presence of BBOA, someone would argue you can see a BBOA in summer and it could be more questionable the fact that you are identifying two types of BBOA. A few lines supporting these two BBOA profiles are suggested.

Thanks for the reviewer's insightful suggestion and we have rephrased this entire paragraph clearly in the revised version. The responses to the comment are separated into four parts as follows.

The confirmation of the presence of BBOA in this study was mainly on the basis of the $m/z$ 60 signals or $f_{60}$ values in the HRMS. According to the previous AMS studies (Cubison et al., 2011; Zhou et al., 2017), if the $f_{60}$ values was significantly larger than ~ 0.3% (a typical value that has been widely used as a background level in air masses not impacted by active open biomass burning), the aerosol was generally regarded as the presence of BBOA. In this study, the $f_{60}$ values in BBOA and agBBOA were 0.51% and 0.46%, respectively, demonstrating the presence of biomass burning related OA factors at WLG site.

In addition, biomass burning activeties in the QTP regions and its arrounding areas were quite common due to the widely usage of wood, grass, dung, and incense for residential cooking or worship at temple. More important, these biomass burning activeties occurred during all seasons rather than just winter or autumn seasons, so it was not strange to identify biomass burning related OA factors in summer season in the QTP regions.

The identification of the two types of BBOA in this study was not only based on the optimal selection of four-factor solution for PMF analysis, but also consist with the fact that biomass burning OAs would have different oxidation degrees when the emissions transported from surrounding areas to WLG site under different oxidation conditions and transport distances. Besides, similar OA source apportionment of two BBOA components with different oxidation degrees have also been resolved in previous studies, e.g., an additional oxygenated biomass-burning-influenced organic aerosol (OOA2-BBOA or OOA-BB) in the Paris metropolitan area (Crippa et al., 2013), urban Nanjing (Zhang et al., 2015) and Mt. Yulong (Zheng et al., 2017), respectively, besides the relatively fresh BBOA component.

Although the mass spectrum of agBBOA in this study looked like a semi-volatile OA, the time series of agBBOA was correlated tightly with $C_2H_4O_2^+$ ($R^2 = 0.79$). In addition, the mass spectrum of agBBOA was also resembled well with that of BBOA at QOMS ($R^2 = 0.954$). Therefore, on the basis of these evidences, the named oxygenated biomass-burning-influenced OA factor was reasonable in this study.

"Two biomass burning related OA factors, a relatively fresh biomass burning OA (BBOA) and an aged biomass burning OA (agBBOA), with distinctly different oxidation degrees were also found in this study. Although the $m/z$ 44 signals were still the highest peaks for both the two factors, the $m/z$ 60 signals, which were generally regarded as well-known tracers for biomass burning emissions (Alfarra et al., 2007), were also obvious in both HRMS. The fractions of the signals at $m/z$ 60 ($f_{60}$) in their HRMS were 0.51 and 0.46%, respectively, which were significantly higher than the typical value of 0.3% that has been widely used as a background level in air masses not impacted by active open biomass burning in previous studies (Cubison et al., 2011; Zhou et al., 2017), demonstrating the presence of biomass burning related OA factors at WLG site. As shown in the Fig. 5, the time series of agBBOA correlated tightly with $C_2H_4O_2^+$ ($R^2 = 0.79$) and sulfate ($R^2 = 0.47$), while BBOA correlated slightly weak with $C_2H_4O_2^+$ ($R^2 = 0.47$) and potassium ($R^2 = 0.30$), respectively. The time series of agBBOA also correlated well with $C_xH_yO_1^+$ and $C_xH_yO_2^+$ ions, while BBOA correlated well with $C_xH_y^+$ and $C_xH_yO_1^+$ (Fig. S10). In addition, both the mass spectra of the two biomass burning related OA factors resembled well with that of BBOA at QOMS ($R^2$ of 0.886 and 0.954, respectively; Fig. S11; Zhang et al., 2018), whereas correlated moderately ($R^2 = 0.39$–0.59) with other standard BBOA mass spectra at other sites around the world (Aiken et al., 2009; Mohr et al., 2012). The agBBOA mass spectrum in this study correlated tightly ($R^2 = 0.914$) with the less oxidized oxygenated OA (LOOOA) identified at Nam Co station (Fig. S11; Xu et al., 2018). All these comparisons and correlation analysis further verified the reasonable source apportionment of OA in this study, namely there were two biomass burning related OAs at WLG, as a result of the different oxidation degrees of biomass burning emissions transported from surrounding areas to WLG site (see Sect. 3.4 for details). Similar OA source apportionment of two BBOA components with different oxidation degrees have also been resolved in previous studies, e.g., an additional oxygenated biomass-burning-influenced organic aerosol (OOA$_2$-BBOA or OOA-BB) in the Paris metropolitan area (Crippa et al., 2013), urban Nanjing (Zhang et al., 2015) and Mt. Yulong (Zheng et al., 2017), respectively, besides the relatively fresh BBOA component."

(3) The authors can also add, maybe to the supplement, more details on the way they selected the four factor solution, information about the Q/Qexp values, residuals, etc.

Thanks for the reviewer's insightful suggestion. We have added a key diagnostic plot to the supplement information, including the information about the $Q/Q_{exp}$ as a function of factor number and $f$Peak, fractions of OA factors vs. $f$Peak, correlations among PMF factors, residuals or scaled residuals for each m/z and time, etc, which would help to understand the way we selected the four factor solution. The description of the figure is added in both the revised manuscript and supplement information, respectively.

[revised manuscript text omitted]

---

## Editor Decision (ED1)

I would like to thank you for the detailed answers to the referees' comments.
After a few technical corrections (here attached), the manuscript can be accepted for publication.

Technical corrections:

Use nitrogen oxides instead of nitric oxides to define NOx
Replace "with diameter less than 10 μm" with "with aerodynamic diameter less than 10 μm" (I assume this is aerodynamic diameter…)
Add ratios in: "higher O/C and OM/OC ratios during"
Add a verb in "Besides the significant impacts by anthropogenic emissions from the northwestern China or Indian subcontinent, air pollutants to the northeastern QTP could also come(?) from the central Eurasian continent (Xue et al., 2013)."
Correct as follows: "Studies focusing on the atmospheric aerosol chemical compositions in the northeastern QTP using the high-time-resolution real-time measurements are still relatively rare until now."
Correct "while BBOA correlated slightly weak with…" . I assume you mean correlated slightly with…" or "correlated weakly with…".
Rephrase as: "…QTP except a study reporting measurements using Aerodyne aerosol chemical speciation monitor (ACSM) at Menyuan (Du et al., 2015)."

---

## Author Response (AR2)

**Chemical characterization and sources of submicron aerosols in the northeastern Qinghai-Tibet Plateau: insights from high-resolution mass spectrometry**

**Xinghua Zhang et al.**

We appreciate the Co-Editor for your technical comments to our manuscript. The manuscript has been revised accordingly and our point-by-point responses are presented below. The comments are in black, followed by responses in blue and revised manuscript in red with changes marked by underline.

**Technical Corrections:**

(1) Use nitrogen oxides instead of nitric oxides to define NOx

Agree.

"Nitrate, oxidized from the nitrogen oxides ($NO_x$), was also an important component..."

(2) Replace "with diameter less than 10 μm" with "with aerodynamic diameter less than 10 μm" (I assume this is aerodynamic diameter…)

Corrected.

"...measurements of $PM_{2.5}$ (particulate matter with aerodynamic diameter less than 2.5 μm)"

"...$PM_{10}$ (particulate matter with aerodynamic diameter less than 10 μm)"

(3) Add ratios in: "higher O/C and OM/OC ratios during"

Agree.

"The relatively higher O/C and OM/OC ratios during afternoon..."

(4) Add a verb in "Besides the significant impacts by anthropogenic emissions from the northwestern China or Indian subcontinent, air pollutants to the northeastern QTP could also come(?) from the central Eurasian continent (Xue et al., 2013)."

Agree.

"...air pollutants to the northeastern QTP could also come from the central Eurasian continent..."

(5) Correct as follows: "Studies focusing on the atmospheric aerosol chemical compositions in the northeastern QTP using the high-time-resolution real-time measurements are still relatively rare until now."

Corrected.

"Studies focusing on the atmospheric aerosol chemical compositions in the northeastern QTP..."

(6) Correct "while BBOA correlated slightly weak with…" . I assume you mean correlated slightly with…" or "correlated weakly with…".

Corrected.

"...while BBOA correlated slightly with..."

(7) Rephrase as: "…QTP except a study reporting measurements using Aerodyne aerosol chemical speciation monitor (ACSM) at Menyuan (Du et al., 2015)."

Corrected.

"...conducted in the northern QTP except a study reporting measurement using..."